# The Value of Information in Human-AI Decision-making

**Ziyang Guo**
Northwestern University
`ziyang.guo@northwestern.edu`

**Yifan Wu**
Microsoft Research
`yifan.wu2357@gmail.com`

**Jason Hartline**
Northwestern University
`hartline@northwestern.edu`

**Jessica Hullman**
Northwestern University
`jhullman@northwestern.edu`

## ABSTRACT

Multiple agents are increasingly combined to make decisions with the expectation of achieving *complementary performance*, where the decisions they make together outperform those made individually. However, knowing how to improve the performance of collaborating agents requires knowing what information and strategies each agent employs. With a focus on human-AI pairings, we contribute a decision-theoretic framework for characterizing the value of information. By defining complementary information, our approach identifies opportunities for agents to better exploit available information in AI-assisted decision workflows. We present a novel explanation technique (ILIV-SHAP) that adapts SHAP explanations to highlight human-complementing information. We validate the effectiveness of our framework and ILIV-SHAP through a study of human-AI decision-making, and demonstrate the framework on examples from chest X-ray diagnosis and deepfake detection. We find that presenting ILIV-SHAP with AI predictions leads to reliably greater reductions in error over non-AI assisted decisions more than vanilla SHAP[1].

## 1 INTRODUCTION

As the performance of artificial intelligence (AI) models continues to improve across domains, workflows in which human experts and AI models are paired for decision-making are sought in medicine, finance, and law, among others. Statistical models can often exceed the accuracy of human experts on average (Ægisdóttir et al., 2006; Grove et al., 2000; Meehl, 1954). However, whenever humans have access to additional information over an AI model, there is potential to achieve *complementary performance* by pairing the two, i.e., better performance than either the human or AI alone. For example, a physician may have access to information that is not captured in structured health records (Alur et al., 2024).

Many empirical studies, however, have found that human-AI teams underperform the AI alone (Buçinca et al., 2020; Bussone et al., 2015; Green & Chen, 2019; Jacobs et al., 2021; Lai & Tan, 2019; Vaccaro & Waldo, 2019; Kononenko, 2001). Two sources of ambiguity complicate such results. One concerns the role of measurement: performance is often scored against post-hoc decision accuracy (Passi & Vorvoreanu, 2022) rather than accounting for the best achievable performance given information available at the time of the decision (Kleinberg et al., 2015; Guo et al., 2024; Rambachan, 2024). Additionally, it often remains unclear how agents differed in their information access or use, making it difficult to design interventions to improve these aspects.

Imagine one could identify information complementarities that can be exploited, such as when one of the agents has access to information not contained in the other's judgments, or has not fully integrated contextually-available information (e.g., instance features) in their judgments. This would motivate interventions to improve decision-making. For example, if we can identify how much complementary

---

[1]The code to calculate the main quantities in our framework and reproduce the experimental results is available at `https://github.com/Guoziyang27/decision_infovalue`.

information AI predictions provide over human judgments, we can use this knowledge to guide model selection or motivate further data collection to improve the model. Conversely, finding evidence that model predictions contain decision-relevant information that humans do not exploit can motivate the design of explanations communicating complementary information.

We contribute a decision-theoretic framework for characterizing the value of information available in an AI-assisted decision workflow. In our framework, information is considered valuable to a decision-maker to the extent that it is possible, in theory, to incorporate it into their decisions to improve performance. Specifically, our approach analyzes the expected marginal payoff gain from best case (Bayes rational) use of additional information over best case use of the information already encoded in agent decisions. The rational framework allows us to upperbound the expected payoff that is achievable by any strategies in the same experiment, and identifies sub-optimality in agent use of information by comparing to rational behavior. The upper bound our framework estimates holds regardless of the human's decision-making process, which may deviate from rationality, given a specified decision problem. Further, our methods can be used even when the decision problem definition is ambiguous, by using a robustness analysis over all possible proper scoring rules to identify the upper bound of performance in the worst case.

We introduce two metrics for evaluating information value in human-AI collaboration. The first— global human-complementary information value—calculates the value of a new piece of information to an agent over all of its possible realizations among all instances. The second—instance-level human-complementary information value—identifies opportunities for decision-makers to better use instance-level information such as specific AI predictions. Applying the second metric, we derive a new explanation technique (ILIV-SHAP) that reveals how data features influence the value of complementary information for an individual prediction.

To evaluate these tools, we contribute the results of a crowdsourced between-subjects experiment in which humans make decisions with and without AI models with varying human-complementary information and different explanation approaches. We find that an AI model with more human-complementary information leads to greater improvements in human-AI team performance over the human-alone baseline. We also find that adding an ILIV-SHAP explanation to a traditional SHAP explanation leads to greater improvements in human-AI performance over the human-alone baseline than only the SHAP explanation or no explanation. We demonstrate further uses of the framework in real-world decision-making tasks, including chest X-ray diagnosis (Rajpurkar et al., 2018; Johnson et al., 2019) and deepfake detection (Dolhansky et al., 2020; Groh et al., 2022)[2].

## 2 RELATED WORK

**Human-AI complementarity.** Many empirical studies of human-AI collaboration focus on AI-assisted human decision-making for legal, ethical, or safety reasons (Bo et al., 2021; Boskemper et al., 2022; Bondi et al., 2022; Schemmer et al., 2022). However, a recent meta-analysis by Vaccaro et al. (2024) finds that, on average, human–AI teams perform worse than the better of the two agents alone. In response, a growing body of work seeks to evaluate and enhance complementarity in human–AI systems (Bansal et al., 2021b; 2019; 2021a; Wilder et al., 2021; Rastogi et al., 2023; Mozannar et al., 2024b). The present work differs from much of these prior works by approaching human-AI complementarity from the perspective of information value and use, including asking whether the human and AI decisions provide additional information that is not used by the other.

**Evaluation of human decision-making with machine learning.** Our work contributes methods for evaluating the decisions of human-AI teams (Kleinberg et al., 2015; 2018; Lakkaraju et al., 2017; Mullainathan & Obermeyer, 2022; Rambachan, 2024; Guo et al., 2024; Ben-Michael et al., 2024; Shreekumar, 2025). Kleinberg et al. (2015) proposed that evaluations of human-AI collaboration should be based on the information that is available at the time of the decision. According to this view, our work defines Bayesian best-attainable-performance benchmarks similar to several prior works (Guo et al., 2024; Wu et al., 2023; Agrawal et al., 2020; Fudenberg et al., 2022). Closest to our work, Guo et al. (2024) model the expected performance of a rational Bayesian agent faced with deciding between the human and AI recommendations as the theoretical upper bound on the expected

---

[2]We also include an observational study on the dataset from Vodrahalli et al. (2022) which identified the AI model with more human-complementary information helps human-AI teams achieve greater improvements in performance over the human-alone baseline. See Appendix D.

performance of any human-AI team. This benchmark provides a basis for identifying exploitable information within a decision problem.

**Complementarity by design.** Some approaches focus on automating the decision pipeline by explicitly incorporating human expertise in developing machine learning models or human-AI collaboration pipeline, such as by learning to defer (Mozannar et al., 2024a; Madras et al., 2018; Raghu et al., 2019; Keswani et al., 2022; 2021; Okati et al., 2021; Chen et al., 2022). Corvelo Benz & Rodriguez (2023) propose multicalibration over human and AI model confidence information to guarantee the existence of an optimal monotonic decision rule. Other approaches exploit information asymmetry (i.e., cases where humans have additional contextual knowledge) and offer principled methods with provable guarantees to improve team decision performance (Straitouri et al., 2023; De Toni et al., 2024; Arnaiz-Rodriguez et al., 2025). Alur et al. (2024) propose a framework to incorporate human decisions into machine learning algorithms when the state is indistinguishable from the algorithm alone but can be discriminated by humans. Our work also concerns information asymmetry, but provides an interpretable analytical framework for quantifying the information value of all available signals and agent decisions in human–AI decision tasks, enabling the design of information-based interventions.

## 3 FRAMEWORK

Our framework takes as input a decision problem associated with an information model, including decisions from one or more agents. It outputs the value of information of available signals to the agents, conditioning on the existing information in their decisions. Understanding the possible value of a signal to a decision-maker requires defining the best attainable decision performance with that signal. We therefore adopt a Bayesian decision theoretic framework.

Our framework provides two separate functions to quantify the value of information: one globally across the data-generating process, and one locally in a realization.

**Decision Problem.** A decision problem consists of three key elements. We illustrate with an example of a weather decision.

- A payoff-relevant state $\omega$ from a space $\mathbf{\Omega}$. For example, $\omega \in \mathbf{\Omega} = \{0, 1\} = \{\text{no rain}, \text{rain}\}$.

- A decision $d$ from the decision space $\mathbf{D}$ characterizing the decision-maker (DM)'s choice. For example, $d \in \mathbf{D} = \{0, 1\} = \{\text{not take umbrella}, \text{take umbrella}\}$.

- A payoff function $S : \mathbf{D} \times \mathbf{\Omega} \to \mathbb{R}$, used to assess the quality of a decision given a realization of the state. For example, $S(d = 0, \omega = 0) = 0, S(d = 0, \omega = 1) = -100, S(d = 1, \omega = 0) = -50, S(d = 1, \omega = 1) = 0$, which punishes the DM for wrongly taking or not taking the umbrella.

**Information Model.** We cast the information available to a DM, including any available agent decisions, as a set of signals defined within an information model. Following the definition of an information model in Blackwell et al. (1951), the information model can be represented by a *data-generating model* with a set of *signals*.

- *Signals*. There are $n$ "basic signals" represented as random variables $\Sigma_1, \ldots, \Sigma_n$, from the signal spaces $\mathbf{\Sigma}_1, \ldots, \mathbf{\Sigma}_n$. Basic signals represent information available to a decision-maker as they decide, e.g., $\mathbf{\Sigma}_1 = \Delta\mathbf{\Omega}$ for a probabilistic prediction of raining, $\mathbf{\Sigma}_2 = \{\text{cloudy}, \text{not cloudy}\}$, $\mathbf{\Sigma}_3 = \{0, \ldots, 100\}$ for temperature in Celsius, $\mathbf{\Sigma}_4 = \mathbf{D}$ for human decisions $D^H$ on taking umbrella or not, etc. The decision-maker observes a signal, which is a subset of the basic signals, $V \subseteq 2^{\{\Sigma_1, \ldots, \Sigma_n\}}$. The combination of two signals $V_1$ and $V_2$ takes the set union $V = V_1 \cup V_2$. In the standard human-AI decision workflow where a human makes an independent judgment before consulting the AI, all basic signals are $\{x, D^H, D^{AI}\}$–the features of the instance, the human's initial decisions and the AI's predictions.

- *Data-generating process*. Decision benchmarks are defined relative to a specific data-generating process, a joint distribution $\pi \in \Delta(\mathbf{\Sigma}_1 \times \ldots \times \mathbf{\Sigma}_n \times \mathbf{\Omega})$ over the basic signals and the payoff-relevant state. $\pi$ can be viewed as the combination of two distributions: the prior distribution of the state $\Pr[\omega]$ and the signal-generating distribution $\Pr[v|\omega]$ defining the conditional distribution of signals. To represent the best-attainable performance of observing a subset $V$ of the $n$ basic signals, we use the Bayesian posterior belief upon receiving a signal $V = v$ as

$$\pi(\omega|v) := \Pr[\omega|v] = \frac{\pi(v, \omega)}{\pi(v)}$$

where $\pi(v, \omega)$ denotes the marginal probability of the signal realized to be $v$ and the state being $\omega$ with expectation over other signals, and $\pi(\omega)$ denotes the prior $\Pr[\omega]$. Throughout the paper, we use capital letters to denote a random variable of signal, e.g., $V$, and use little letters to denote a realization of signal, e.g., $v$.

**Information value.** Our framework quantifies the value of information in a signal $V$ as the expected payoff improvement of an idealized agent who has access to $V$ in addition to some baseline information set. This corresponds to a rational Bayesian DM who knows the prior probability of the state and conditional distribution of signals (i.e., the data-generating process), observes a signal realization, updates their prior to arrive at posterior beliefs, and then chooses a decision to maximize their expected payoff given their posterior belief. Formally, given a decision task with payoff function $S$ and an information model $\pi$, the rational DM's expected payoff given a (set of) signal(s) $V$ is

$$\mathrm{R}(V) := \mathbf{E}_{(v, \omega) \sim \pi}[S(d^r(v), \omega)] \tag{1}$$

where $d^r(\cdot) : \mathbf{V} \to \mathbf{D}$ denotes the decision rule adopted by the rational DM.

$$d^r(v) = \arg\max_{d \in \mathbf{D}} \mathbf{E}_{\omega \sim \pi(\omega|v)}[S(d, \omega)] \tag{2}$$

We further characterize the maximum expected payoff that can be achieved with no information. This can be used as the baseline to quantify the information value of $V$ as the payoff improvement of $V$ over. We use $\emptyset$ to represent a null signal and $\mathrm{R}(\emptyset)$ represents the expected payoff of a Bayesian rational DM who only uses the prior distribution to make decisions. In this case, the Bayesian rational DM takes the best fixed action under the prior, and their expected payoff is:

$$\mathrm{R}(\emptyset) := \max_{d \in \mathbf{D}} \mathbf{E}_{\omega \sim \pi}[S(d, \omega)] \tag{3}$$

**Definition 3.1.** Given a decision task with payoff function $S$ and an information model $\pi$, the information value of $V$ is defined as

$$\mathrm{IV}(V) := \mathrm{R}(V) - \mathrm{R}(\emptyset) \tag{4}$$

The full information value in a human-AI decision task is the information value of the set of all basic signals. For example, in the weather decision example, $\mathrm{IV}(\{\Sigma_1, \Sigma_2, \Sigma_3, \Sigma_4\})$ represents the full information value of the probabilistic prediction of rain, the cloudiness, the temperature, and the human decisions. This defines the upper bound of the information value of any signal, including the agent-complementary information value and instance-level agent-complementary information value that we will define in the following sections.

### 3.1 Agent-Complementary Information Value

With the above definitions, it is possible to measure the best case additional value that new signals can provide over the information already captured by an agent's decisions. Here, *agent* may refer to a human, an AI system, or a human–AI team. The intuition behind our approach is that any information that is used by decision-makers should eventually reveal itself through variation in their decisions. Definition 3.2 captures how much complementary information value is offered by a signal over the agent decisions in expectation over the data-generating process. For example, in the weather decision task, this indicates how much the payoff from the decision-maker's choice of whether to take the umbrella or not can be further improved by incorporating the temperature in their decision rule.

**Definition 3.2.** Given a decision task with payoff function $S$ and an information model $\pi$, we define the agent-complementary information value (ACIV) of $V$ on agent decisions $D^b$ as

$$\mathrm{ACIV}(V; D^b) := \mathrm{R}(D^b \cup V) - \mathrm{R}(D^b) \tag{5}$$

If the ACIV of a signal $V$ is small relative to the baseline (3), this means either that the information value of $V$ to the decision problem is low (e.g., it is not correlated with $\omega$), or that the agent has already exploited the information in $V$ (e.g., the agent relies on $V$ or equivalent information to make their decisions such that their decisions correlate with $\omega$ in the same way as $V$ correlates with $\omega$). If, however, the ACIV of $V$ is large relative to the baseline, then at least in theory, the agent can improve their payoff by incorporating $V$ in their decision making.

Furthermore, ACIV can reveal complementary information between different types of agents. For instance, if we view AI predictions as $V$ and treat available human decisions as the agent signal $D^b$, a large ACIV indicates that AI predictions add considerable value beyond what humans alone achieve. In the reverse scenario, if human decisions serve as $V$ and AI predictions are $D^b$, we can measure how much humans can contribute over the information captured in the AI predictions.

Of course, we are unlikely to observe identical realizations of the signals in continuous-valued or high-dimensional data, such as images or text. A natural relaxation is to consider the realizations that are sufficiently "similar" for the performance of the decision-maker. Algorithm 1 learns the posterior distribution of the payoff-relevant state from the signal and then uses that prediction as the probability distribution of the payoff-relevant state to choose the optimal action. This approach can be readily extended to the case where the signal is more complex, such as human decisions in the form of freeform text (e.g., radiology reports) or explanations of the AI predictions in the form of images (e.g., saliency map). [3]

---

**Algorithm 1** A method to calculate the ACIV of $V$.

1: **Input:** Observed realizations $\{v_i, d_i^b, \omega_i\}_{i=1}^n$, a predictive algorithm $\mathcal{A}$, payoff function $S$, decision space $\mathbf{D}$, and state space $\mathbf{\Omega}$.
2: **Output:** $\text{ACIV}(V; D^b)$
3: $\hat{a} \leftarrow \mathcal{A}(\{v_i, d_i^b, \omega_i\}_{i=1}^n)$
4: $\hat{a}^b \leftarrow \mathcal{A}(\{d_i^b, \omega_i\}_{i=1}^n)$
5: **for** $i = 1$ to $n$ **do**
6: $\quad \hat{p}_i \leftarrow \hat{a}(v_i, d_i^b)$
7: $\quad \hat{d}_i^r \leftarrow \arg\max_{d \in \mathbf{D}} \mathbf{E}_{\omega \sim \hat{p}_i}[S(d, \omega)]$
8: $\quad \hat{p}_i^b \leftarrow \hat{a}^b(d_i^b)$
9: $\quad \hat{d}_i^{rb} \leftarrow \arg\max_{d \in \mathbf{D}} \mathbf{E}_{\omega \sim \hat{p}_i^b}[S(d, \omega)]$
10: $\quad s_i \leftarrow S(\hat{d}_i^r, \omega_i) - S(\hat{d}_i^{rb}, \omega_i)$
11: **end for**
12: **return** $\frac{1}{n} \sum_{i=1}^n s_i$

---

## 3.2 Instance-level Agent-complementary Information Value

ACIV quantifies the value of the decision-relevant information in a signal $V$ across the distribution of all possible realizations defined by the data-generating model. To provide analogous instance-level quantification of information value, we define Instance-Level agent-complementary Information Value (ILIV) to quantify the value of the decision-relevant information encoded by a single realization of the signal. This finer-grained view makes it possible to analyze how much an agent can benefit in theory from better incorporating instance-level information in their decision. For example, in the weather decision task, this indicates how much the payoff of the decision-maker can be improved by knowing the temperature is 21°C.

Given a realization of signal $v = \{\sigma_{j_1}, \ldots, \sigma_{j_k}\}$, we want to know the maximum expected payoff gain from the access to $v$ on the instances where $v$ is realized over the existing information encoded in agent decisions. Intuitively, this captures how much "room" there is for a specific signal to be better used. Formally, given a decision task with payoff function $S$ and information model $\pi$, the expected payoff of the rational DM given signal $V = v'$ on instances with the signal realization $V = v$ is

$$\text{r}^v(v') := \mathbf{E}_{\omega \sim \pi(\omega|v)}[S(d^r(v'), \omega)] \tag{6}$$

where $d^r(v')$ is the Bayesian optimal decision on receiving $v'$ as defined in Equation (2). Note that we allow $v' \neq v$, i.e. the signal $v'$ observed by the rational DM can be different from the actual realization $v$. The expected payoff of a rational DM who is misinformed is guaranteed to be lower than is correctly informed, i.e., $\text{r}^v(v') \leq \text{r}^v(v)$ for any $v'$. This notion of flexibility allows us to consider the information value of a counterfactual realization of a signal. For example, in the weather decision task, this notation is able to describe how much payoff can change when the decision maker is misinformed that the temperature is 18°C when the temperature is actually 21°C. We use this in designing the explanation of ILIV-SHAP in the next section.

---

[3]Note that the algorithm $(\hat{a}, \hat{a}^b)$ should be cross-validated to avoid overfitting to the observed data and also be evaluated for calibration error since the rational DM treats it as the Bayesian posterior. We provide a sensitivity analysis in Appendix I where we compare the ACIV estimated under the rational belief estimator using linear regression (LR), gradient boosting methods (GBM), and neural networks (NN).

If we consider the agent decisions in addition to the realization $v$, the rational DM's expected payoff on instances where $V = v$ can be written as

$$\mathrm{r}^v(v'; D^b) := \mathbf{E}_{(d^b, \omega) \sim \pi(d^b, \omega | v)}[S(d^r(v' \cup d^b), \omega)] \tag{7}$$

**Definition 3.3.** Given a decision task with payoff function $S$ and an information model $\pi$, we define the instance-level agent-complementary information value (ILIV) of signal realization $V = v'$ on instances where $V = v$ as:

$$\mathrm{ILIV}^v(v'; D^b) := \mathrm{r}^v(v'; D^b) - \mathrm{r}^v(\emptyset; D^b). \tag{8}$$

where $\mathrm{r}^v(\emptyset; D^b)$ represents the expected rational payoff on instances of $V = v$, where the rational DM only knows the agent decisions without knowing any realizations of $V$. ILIV maximizes when the signal does not misinform, i.e., $\mathrm{ILIV}^v(v; D^b) \geq \mathrm{ILIV}^v(v'; D^b)$, for $v' \in \mathbf{V}$.

## 4 INFORMATION-BASED EXPLANATION

We define an *information-based* explanation (ILIV-SHAP) to communicate where the AI prediction offers complementary information over the agent decisions. Traditional saliency-based explanations communicate the average contribution of each feature to a prediction over the baseline (prior) prediction, while ILIV-SHAP communicates the average contribution of each feature to the agent-complementary information value contained in the prediction. Algorithm 3 estimates the ILIV given a set of observed signals, agent decisions, and states.

Specifically, suppose a model $f$ that, for example, takes as input $m$ features and outputs a real number. Given an instance $\mathbf{x} = (x_1, \ldots, x_m)$, the importance of one feature $x_i$ to the model output $f(\mathbf{x})$ is encoded by the expected difference of model outputs when $x_i$ is marginalized out. This is quantified by $f(\mathbf{x}) - \mathbf{E}[f(X)|X_{-i} = \mathbf{x}_{-i}]$, where $X_{-i}$ denotes all features except $X_i$. Considering the interaction between features, SHAP (Lundberg & Lee, 2017) uses the Shapley value to quantify the importance scores averaged on different combinations of features:

$$\phi_i(f, \mathbf{x}) = \sum_{\mathbf{x}' \subseteq \mathbf{x}} \frac{|\mathbf{x}'|!(m - |\mathbf{x}'| - 1)!}{m!} [g_f(\mathbf{x}') - g_f(\mathbf{x}' \backslash x_i)]$$

where $g_f(\mathbf{x}')$ denotes the expected output conditioned on $\mathbf{x}'$, i.e., $\mathbf{E}[f(X)|X' = \mathbf{x}']$ for any $\mathbf{x}' \subseteq \mathbf{x}$ where $\mathbf{x}'$ is the features that are not marginalized out.

**Definition 4.1** (ILIV-SHAP). Given a model $f$ and data features $\mathbf{x} = (x_1, \ldots, x_m)$, the importance score of the $i$-th feature by ILIV-SHAP is

$$\phi_i^{\mathrm{ILIV}}(f, \mathbf{x}) = \sum_{\mathbf{x}' \subseteq \mathbf{x}} \frac{|\mathbf{x}'|!(m - |\mathbf{x}'| - 1)!}{m!} [\mathrm{ILIV}^{f(\mathbf{x})}(g_f(\mathbf{x}'); D^b) - \mathrm{ILIV}^{f(\mathbf{x})}(g_f(\mathbf{x}' \backslash x_i); D^b)]$$

where $\mathrm{ILIV}^{f(\mathbf{x})}(g_f(\mathbf{x}'); D^b)$ denotes a counterfactual evaluation of ILIV, which quantifies the expected payoff gain from additionally knowing $g_f(\mathbf{x}')$ over $D^b$ on the instances where the actual prediction is $f(\mathbf{x})$.

ILIV-SHAP shares the same properties as SHAP (Lundberg & Lee, 2017) by similarly constructing the importance scores. For example, ILIV-SHAP satisfies the critical *efficiency* axiom, i.e., the sum of the importance scores equals the information value of the model output, and the *symmetry* axiom, i.e., the importance scores are the same for any two features that contribute the same amount to the information value of the model output.

Sample-based methods for approximating SHAP values[4] can also be applied to ILIV-SHAP, such as permutation sampling (Strumbelj & Kononenko, 2010), Kernel SHAP (Lundberg & Lee, 2017), and Partition SHAP (Lundberg et al., 2018). Because ILIV is expected to be non-decreasing in the number of features included in $g_f(\mathbf{x}')$—more features mean more information—sample-based methods can achieve better stability across the permutations on ILIV-SHAP than on SHAP.

## 5 EXPERIMENT

We use a preregistered between-subjects online experiment to answer two questions:

---

[4]We note that ILIV-SHAP does not resolve the foundational issues of SHAP (Huang & Marques-Silva, 2024), but just as a demonstration on how the information value such as ILIV can be used in explanations.

| Feature name | Value | Impact on AI prediction |
|---|---|---|
| Feature Y | 100 | -$6.2K |
| # of Fireplaces | 0 | -$13.3K |
| Year Built | 1900 | -$7.5K |
| Car Capacity in Garage | 2.0 | -$1.1K |
| Feature X | 380 | +$4.8K |
| Above Grade Living Area | 1627sq ft | +$1K |

Model prediction: $134.9K
Your previous guess: $160K
Highlighted features are where the model gets significant amount of human-complementary information.

| Feature name | Value | Impact on AI prediction |
|---|---|---|
| Year Built | 1952 | -$11.5K |
| Above Grade Living Area | 1044sq ft | -$8.4K |
| # of Fireplaces | 0 | -$10.3K |
| Car Capacity in Garage | 2.0 | -$0.5K |
| Feature X | 1460 | $0K |
| Feature Y | 100 | $0K |

Model prediction: $112.9K
Your previous guess: $160K
Highlighted features are where the model gets significant amount of human-complementary information.

(a) AI1 with ILIV-SHAP and SHAP explanation.  (b) AI2 with ILIV-SHAP and SHAP explanation.

Figure 1: The screenshot of the interface for ILIV-SHAP and SHAP explanations. The third column is the SHAP value and the highlights are based on the ILIV-SHAP value. Because AI2 does not have access to Feature X/Y, both their ILIV-SHAP and SHAP values are zero for AI2.

1. Can ACIV identify which AI model will result in the best human-AI decision-making?
2. Does ILIV-SHAP improve human-AI decision-making over SHAP alone?

Our experiment assigns participants to one of two AI models with varying ex-ante ACIV {AI1=high ACIV, AI2=low ACIV}, and one of three explanation conditions {ILIV-SHAP and SHAP, SHAP, No Explanation}. We hypothesize that the higher ACIV model will lead to better human-AI decisions, and that presenting ILIV-SHAP explanations for this model will lead to better human-AI decisions than the other explanation types.

**Task and data.** We study AI-assisted house price prediction, following other recent crowdsourced studies (e.g., Chiang & Yin (2021); Hemmer et al. (2022); Poursabzi-Sangdeh et al. (2021); Holstein et al. (2023)). This task does not require any domain knowledge, making it suitable for a broad participant population. Additionally, the task is complex enough to benefit from AI assistance. We use the Ames, Iowa Housing Dataset (De Cock, 2011).

**Participants and procedure.** We recruited 421 US-based participants via the crowdsourcing platform Prolific. Each participant was randomly assigned to 12 houses out of 733 in the Ames Iowa Housing Dataset. Each completed a sequence of 24 trials, where two decisions were elicited for each house: one without AI assistance and one with. In the first 12 trials, participants were asked to predict the house price of the assigned house using six features of the house, including *year built*, the *living area above ground* in square feet, the size of garage in *car capacity*, the number of *fireplaces*, *year remodeled*, and ratings of the house's *material and finish*. We intentionally reduced the interpretability of the *year remodeled* and *material and finish* by relabeling them as "Feature X" and "Feature Y" and rescaling their values. Therefore, the AI model that takes these two features as input is expected to have complementary information over human participants. In the second 12 trials, participants were asked to revise their guesses in the first round after seeing the AI's prediction and explanation. We incentivized participants with a base reward of 3.00 USD and a bonus based on the mean squared error (MSE) between their guesses and the true house price in the 24 trials: $bonus = 3.00 - \frac{MSE}{3 \times 10^9}$.

**AI models and explanations.** Participants were assigned to one of six experimental conditions resulting from crossing the two AI models with the three explanation types. We designed AI1 and AI2 to have varying potential to complement humans: AI1 was trained with the six features and the true house price, with noise added to the six features (which is intended to lower the accuracy of AI1 to be comparable with AI2). On the other hand, AI2 was trained with the four human-interpretable features and the true house price, such that there would be less potential for the predictions to add complementary information over human judgments. We used the same training data for both AI models and ensured that the two AI models achieved similar performance (Figure 2 left), so that model accuracy differences would not confound our results. In the SHAP explanation, we used the SHAP values of the six features to explain the AI's prediction. To generate the ILIV-SHAP explanations, we used independent human decisions we collected on Prolific for the dataset using the same task and features prior to running the official study. We present the exact ILIV-SHAP values

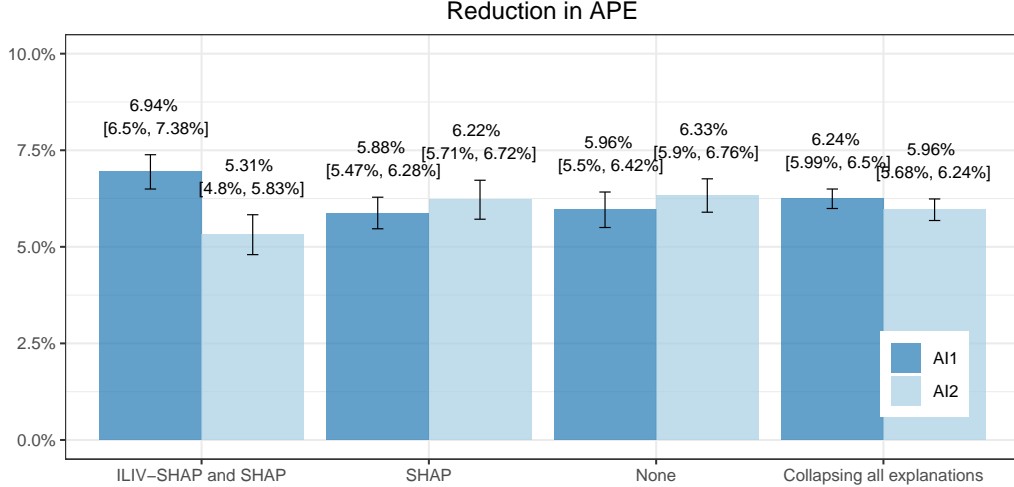

Figure 2: Reduction in the human-AI team's absolute percentage error (APE) with 95% confidence intervals according to posterior predictions of our regression model.

Table 1: The two AI models that are used in the experiment. AI1 has access to all the features, while AI2 has access to only the human-interpretable features.

| | Input features | MAPE | R squared | ACIV (in MSE) | ACIV (in MAPE) |
|---|---|---|---|---|---|
| AI1 | **All** features | 14.30% | 0.81 | $\mathbf{6.5 \times 10^8}$ | **4.61%** |
| AI2 | **Human-interpretable** features | 14.51% | 0.81 | $3.7 \times 10^8$ | 2.00% |

of the six features, which ranked and highlighted the features where the human-complementary information (ILIV) over the AI's prediction exceeds a threshold.[5] See Appendix J for the screenshots of the explanations and instructions.

**Evaluation metrics.** We report the ACIV of the AI models in mean squared error (MSE)[6] and mean absolute percentage error (MAPE). To measure the human alone and human-AI team performance, we fit a preregistered Bayesian hierarchical regression model with weakly informed priors[7] to the percentage error (PE) between the human's prediction and the true house price, $PE = (d - \omega)/\omega$, using R's `brms` package (Bürkner, 2017).

$$PE \sim \text{student\_t}(\mu, \sigma, \nu)$$
$$\mu = AI * explanation + round + (1|participant\_id)$$
$$\log(\sigma) = AI * explanation + round + (1|participant\_id)$$
$$\nu \sim \text{Gamma}(2, 0.1)$$

where *AI* and *explanation* are indicators of the experimental conditions, *round* is an indicator of whether the trial is in the first or second round, and *participant_id* is a unique identifier for each participant. We evaluate human-AI team performance by the reduction in absolute percentage error (APE) over human-alone predictions in the first round, i.e., Reduction in APE = $\mathbf{E}[|PE| \mid round = 1] - \mathbf{E}[|PE| \mid round = 2]$, where the expectation is taken over the Bayesian posterior distribution of the above Bayesian model.

**Results.** Table 1 confirms the expected ranking of AI models by ACIV. AI1, which had access to the two human-uninterpretable features, achieves higher ACIV than AI2, which only had access to the four human-interpretable features. Note the similar predictive performance of the two AIs.

The posterior predictions of human-AI team performance from our regression model also align with expectations based on the complementary information each model provides (Figure 2 far

---

[5]We choose the threshold to highlight the features that can lead to at least $0.25 boost on the bonus to participants, which is translated into $7.5 \times 10^8$ MSE as the threshold for ILIV-SHAP values.

[6]We included MSE since it is a proper scoring rule while MAPE is not.

[7]https://aspredicted.org/jfm6-rrs3.pdf

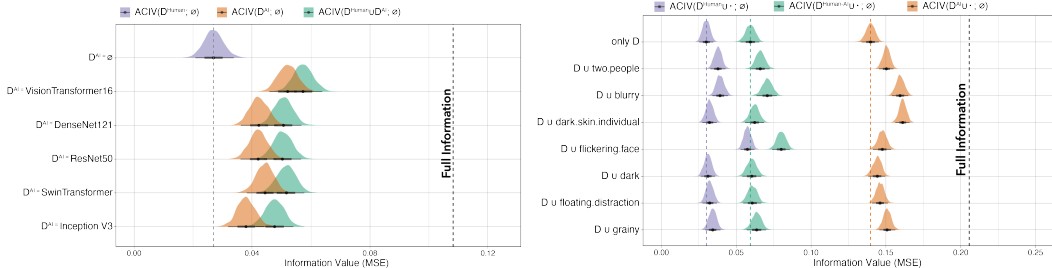

(a) Chest X-ray diagnosis results.     (b) Deepfake detection results

Figure 3: Distributions show information values. For a), we plot ACIV of **radiologist decisions**, different choices of **AI models** and signals that **combine AI predictions and radiologist decisions**. For b), we plot the information value of the combination of video-level features and agent decisions (including **human decisions**, **AI predictions**, and **human-AI teams' decisions**). Full information represents all the information available to the human decision-makers–the radiology images for the chest X-ray diagnosis task and the seven video-level features, human decisions, and AI predictions for the deepfake detection task. See Appendix I for the sensitivity analysis of Algorithm 1.

right). Collapsing across explanation conditions, AI1 (higher ACIV) reduces APE by more than AI2 ($6.24\%[5.99\%, 6.5\%]$ vs $5.96\%[5.68\%, 6.24\%]$ respectively). Access to the ILIV-SHAP + SHAP explanation results in a greater reduction in APE for AI1 than AI2 ($6.94\%[6.50\%, 7.38\%]$ vs $5.31\%[4.80\%, 5.83\%]$ respectively)[8].

Figure 2 also illustrates the effect of the different explanation conditions on human-AI team performance. When the AI has sufficient complementing information (AI1), the ILIV-SHAP + SHAP explanation more effectively reduces APE for the human-AI decisions than the SHAP explanation and baseline (no explanation) ($6.94\%[6.50\%, 7.38\%]$ for ILIV-SHAP, $5.88\%[5.47\%, 6.28\%]$ for SHAP, and $5.96\%[5.50\%, 6.42\%]$ for no explanation).

## 6 DEMONSTRATIONS

### 6.1 CHEST X-RAY DIAGNOSIS

We study a chest X-ray diagnosis task. As agent decisions, $D^{AI}$ and $D^{H}$, we consider five predictive models fine-tuned on the MIMIC-CXR database (Johnson et al., 2019) and radiologists' textual reports recorded in MIMIC-CXR. We use five pretrained image models with the same choices in Irvin et al. (2019). We train the models on 12,228 radiographs, and validate on 6,115 randomly sampled radiographs. For the payoff-relevant state, $\omega \in \mathbf{\Omega} = \{0, 1\}$, we used results from two types of blood tests ('NT-proBNP' and 'troponin' cut by age-specific thresholds). The decision task is formalized as a prediction problem with $\mathbf{D} = \Delta\mathbf{\Omega}$ and $S(d, \omega) = 1 - (d - \omega)^2$.

**Can the AI models complement human judgment?** Figure 3a shows all the models offer complementary information value to the radiologists' reports (**green distributions** improve over the **purple distribution**), and in the other direction, the radiologists' reports offer complementary information value to the models (**green distributions** improve over the **orange distributions**).

**Which AI model offers the most decision-relevant information over human judgments?** Figure 3a shows that VisionTransformer contains slightly higher information value than the other models, and Inception v3 contains slightly lower information value than the other models. We further assess the robustness of VisionTransformer's superiority over the other AI models across many possible payoff functions to test if there is a Blackwell ordering of models in Appendix H and Figure 6.

As shown in this demonstration, doctors may use our framework to learn how much complementary information value the AI models offer over their decisions, and which model offers the most.

### 6.2 DEEPFAKE DETECTION

We study a deepfake detection task (Dolhansky et al., 2020; Groh et al., 2022). We select the model in the Deepfake Detection Challenge (Zhang et al., 2016), with estimated 65% accuracy on holdout data. We use the human decisions and the human-AI team's decisions from Groh et al. (2022), who collected judgments on the videos from n=5,524 participants recruited on Prolific. We use the Brier score as the payoff function, with the binary payoff-related state: $\omega \in \{0, 1\} = \{\text{genuine}, \text{fake}\}$. This

---

[8]See Appendix E for the significant test results

choice differs from Groh et al. (2022)'s use of mean absolute error, but again we prefer the quadratic score because it is a proper scoring rule where truthfully reporting beliefs maximizes the score. Groh et al. (2022) manually label seven video-level features, which we use as binary indicators in place of Algorithm 1 in light of the high dimensional signals: graininess, blurriness, darkness, presence of a flickering face, presence of two people, presence of a floating distraction, and the presence of an individual with dark skin[9]. We show the ACIVs of the seven features over the decisions, the AI predictions, and the human-AI decisions in Figure 3b.

**How much decision-relevant information do agents' decisions offer?**  We first compare the information value of the AI predictions to that of the human decisions.  Figure 3b shows that **AI predictions** provide about $65\%$ of the total possible information value over the no-information baseline, while **human decisions** only provide about $15\%$. We next consider the **human-AI decisions**. Given that the AI predictions contain a significant portion of the total possible information value, we might hope that when participants have access to the AI predictions, their performance will be close to the full information baseline. However, the information value of the **human-AI decisions** only achieves a small proportion of the total possible information value ($30\%$).

**How much additional decision-relevant information do the available features offer over agents' decisions?**  To understand what information might improve human decisions, we assess the ACIVs of different video-level features over different agents. As shown on the fifth row in Figure 3b, the presence of a flickering face offers larger ACIV over human decisions than over AI predictions, meaning that human decisions could improve by a greater amount if they were to incorporate this information. Meanwhile, as shown on the fourth row in Figure 3b, the presence of an individual with dark skin offers larger ACIV over AI predictions than over human decisions, suggesting that humans make greater use of this information. This suggests that the AI and human rely on differing information to make their initial predictions.[10]

## 7    DISCUSSION AND LIMITATIONS

Our decision-theoretic framework quantifies the information value of signals available in a human-AI decision setting over the information value of agent decisions. Importantly, the basis of our framework in Bayesian decision theory does not require that actual (e.g., human) decision-makers achieve Bayesian rational decision-making. Rather, it provides a theoretical basis to support comparisons to human behavior to drive learning (see, e.g., (Guo et al., 2024; Hullman & Gelman, 2021; Wu et al., 2023). This theoretical basis is necessary to establish well-defined benchmarks. We experimentally demonstrate the power of the framework for analyzing and improving human-AI decision-making. Our proposed ILIV-SHAP explanation improves performance over an existing state-of-the-art explanation strategy. This suggests that communicating the value of information that an AI model used to a human facing a decision problem improves the efficacy of explanations over simply explaining how the AI arrived at its prediction. Our work demonstrates a new direction in explanation design grounded in decision theory (e.g., Hullman et al. (2025)).

Our work provides a promising methodological framework for an emerging research agenda around optimally combining agents' information for decisions (e.g., Alur et al. (2023; 2024)). While we focused on human-AI decisions, the framework can be applied to any combination of AI or human agent judgments. Though most of our definitions and analysis are focused on decision tasks with single well-defined payoff function, the framework can be readily extended to more complex decision tasks with an ambiguous set of payoff functions or totally unidentifiable payoff functions. We present a robustness analysis framework in Appendix B to calculate the Blackwell ordering of models over proper scoring rules. A demonstration of the robustness analysis is shown in Appendix H.

The experiment is designed to provide complementarity by construction in order to function as a proof of concept. We believe future work is needed to evaluate the impact of ILIV-SHAP in real deployment of human-AI decision-making workflows, such as where the human or AI might have additional private information over each other.

---

[9]The "dark-skin individual" label reflects a subjective visual attribute. It is used here only for robustness analysis—not for causal interpretation or demographic inference.

[10]For space constraints, we refer to the full descriptions and results in Appendices G and H.

## 8    REPRODUCIBILITY STATEMENT

The main results of our paper are to provide a framework guiding the analysis of complementary information in human-AI decision-making. We provide a Python library in supplementary material to calculate the quantities defined in our framework.

We provide the data and code (which are all put in separate Jupyter notebooks) to reproduce all the results in our paper, including the demonstrations and the empirical study in the main text, and the observational study and robustness analysis in the Appendix.

ACKNOWLEDGEMENTS

Thanks to Advik Shreekumar for his help on using MIMIC-CXR and MIMIC-IV datasets. Thanks to the ICLR reviewers for helpful suggestions (specifically, to include sensitivity analysis and cognitive load).

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

# A   The Combinatorial Nature of the Value of Signals

When decision-makers are provided with multiple signals, the signals have the combinatorial property by nature. Acknowledged by recent works in decision theory and game theory (Chen & Waggoner, 2016), one signal may have no information value by itself, but it can be complementary to other signals to provide information value. For example, two signals $\Sigma_1$ and $\Sigma_2$ are uniformly random bits and the state $\omega = \Sigma_1 \oplus \Sigma_2$, the XOR of $\Sigma_1$ and $\Sigma_2$. In this case, neither of the signals offers information value on its own, but knowing both leads to the maximum payoff. Though we did not explicitly observe the complementation between signals in our survey of human-AI decision-making tasks (see the results of deepfake detection in supplementary materials), we want to note that our framework can be extended to consider this complementation between signals. We use the Shapley value (Shapley, 1953) to interpret the contribution to the ACIV of each basic signal. $\phi$ is the average of the marginal contribution of a signal in every combination with other signals.

$$\phi(V) = \frac{1}{n} \sum_{V' \subseteq \{\Sigma_1, \dots, \Sigma_n\}/V} \binom{(n-|V|)}{|V'|}^{-1} (\mathrm{ACIV}^{\pi,S}(V' \cup V; D^b) - \mathrm{ACIV}^{\pi,S}(V'; D^b)) \quad (9)$$

The following algorithm provides a polynomial-time approximation of the Shapley value of ACIV. Under the assumption of submodularity, it orders the signals the same as the Shapley value.

---

**Algorithm 2** Greedy algorithm for marginal gain of ACIV

---

1: $V^* = \{D^b\}$
2: $\Phi^* = \{\}$
3: **for** $i = 1$ to $n$ **do**
4:    $\phi'_j = \mathrm{ACIV}(\Sigma_j; V^*)$ *for each* $j$
5:    $j^* = \arg\max_{j \ s.t. \ \Sigma_j \notin V^*} \phi'_j$
6:    $\phi_{j^*} = \max_{j \ s.t. \ \Sigma_j \notin V^*} \phi'_j$
7:    *add* $\Sigma_{j^*}$ *to* $V^*$
8:    *add* $\phi_{j^*}$ *to* $\Phi^*$
9: **end for**
10: *output* $\phi_{j^*}$

---

# B   Robust Analysis of Information Order

Our approach assumes a decision problem as input and evaluates agents' decisions and use of information on this problem. However, evaluators may face ambiguity around the appropriate decision problem specification, and in particular, the appropriate scoring rule. In particular, ambiguity can arise in payoff functions; doctors, for example, penalize false negative results differently when diagnosing younger versus older patients (Mclaughlin & Spiess, 2023). Blackwell's comparison of signals (Blackwell et al., 1951) is an ideal tool for addressing ambiguity about the payoff function, as it defines a signal $V_1$ as *more informative* than $V_2$ if $V_1$ has a higher information value on all possible decision problems. We identify this partial order by decomposing the space of decision problems via a basis of proper scoring rules (Li et al., 2022; Kleinberg et al., 2023).

**Definition B.1** (Blackwell Order of Information). A signal $V_1$ is Blackwell more informative than $V_2$ if $V_1$ achieves a higher best-attainable payoff on any decision problems:

$$\mathrm{R}^{\pi,S}(V_1) \geq \mathrm{R}^{\pi,S}(V_2), \forall S$$

where $\mathrm{R}^{\pi,S}(V)$ denotes the expected performance of the rational DM on payoff function $S$ and information structure $\pi$ when observing $V$.

The Blackwell order is evaluated over all possible decision problems, which cannot be tested directly. Fortunately, we only need to test over all proper scoring rules since any decision problem can be represented by an equivalent proper scoring rule, and the space of proper scoring rules can be characterized by a set of V-shaped scoring rules. A V-shaped scoring rule is parameterized by the kink of the piecewise-linear utility function.

**Definition B.2.** (V-shaped scoring rule) A V-shaped scoring rule with kink $\mu \in (0, \frac{1}{2}]$ is defined as

$$S_\mu(d, \omega) = \begin{cases} \frac{1}{2} - \frac{1}{2} \cdot \frac{\omega - \mu}{1 - \mu} & \text{if } d \leq \mu \\ \frac{1}{2} + \frac{1}{2} \cdot \frac{\omega - \mu}{1 - \mu} & \text{else}, \end{cases}$$

When $\mu' \in (\frac{1}{2}, 1)$, the V-shaped scoring rule can be symmetrically defined by $S_{\mu'} = S_{1-\mu'}(1 - y, \omega)$.

Intuitively, the kink $\mu$ represents the threshold belief where the decision-maker switches between two actions. Larger $\mu$ means that the decision-makers will prefer $d = 1$ more. The closer $\mu$ is to 0.5, the more indifferent the decision-maker is to $d = 0$ or $d = 1$.

Proposition B.3 shows that if $V_1$ achieves a higher information value on the basis of V-shaped proper scoring rules than $V_2$, then $V_1$ is Blackwell more informative than $V_2$. Proposition B.3 follows from the fact that any best-responding payoff can be linearly decomposed into the payoff on V-shaped scoring rules.

**Proposition B.3** (Hu & Wu 2024). *If $\forall \mu \in (0, 1)$*

$$R^{\pi, S_\mu}(V_1) \geq R^{\pi, S_\mu}(V_2),$$

*then $V_1$ is Blackwell more informative than $V_2$.*

Extending this to ACIV, $V_1$ offers a higher complementary value than $V_2$ under the Blackwell order if

$$\text{ACIV}^{\pi, S_\mu}(V_1; D^b) \geq \text{ACIV}^{\pi, S_\mu}(V_2; D^b), \forall \mu \in (0, 1)$$

This definition allows us to rank signals (or sets of signals) without needing to commit to a specific payoff function. We present a use case in Appendix G.

## C  ALGORITHM TO ESTIMATE ILIV

---

**Algorithm 3** A method to calculate the ILIV of $v'$ on the instances of $v$.

---

1: **Input:** Observed realizations $\{v_i, d_i^b, \omega_i\}_{i=1}^n$, test counterfactual signal realization $v'$, test signal realization $v$, a predictive algorithm $\mathcal{A}$, payoff function $S$, decision space $\mathbf{D}$, and state space $\mathbf{\Omega}$.
2: **Output:** $\text{ILIV}^v(v'; D^b)$
3: $\hat{a} \leftarrow \mathcal{A}(\{v_i, d_i^b, \omega_i\}_{i=1}^n)$
4: $\hat{a}^b \leftarrow \mathcal{A}(\{d_i^b, \omega_i\}_{i=1}^n)$
5: $\{j_1, \ldots, j_k\} \leftarrow$ the indices of instances where $v_{j_i} = v$ for any $i \in [k]$
6: **for** $i = 1$ to $k$ **do**
7: $\quad \hat{p}_i \leftarrow \hat{a}(v', d_{j_i}^b)$
8: $\quad \hat{d}_i^r \leftarrow \arg\max_{d \in \mathbf{D}} \mathbf{E}_{\omega \sim \hat{p}_i}[S(d, \omega)]$
9: $\quad \hat{p}_i^b \leftarrow \hat{a}^b(d_{j_i}^b)$
10: $\quad \hat{d}_i^{rb} \leftarrow \arg\max_{d \in \mathbf{D}} \mathbf{E}_{\omega \sim \hat{p}_i^b}[S(d, \omega)]$
11: $\quad s_i \leftarrow S(\hat{d}_i^r, \omega_i) - S(\hat{d}_i^{rb}, \omega_i)$
12: **end for**
13: **return** $\frac{1}{k} \sum_{i=1}^k s_i$

---

## D  OBSERVATIONAL STUDY ON THE EFFECT OF ACIV ON HUMAN-AI TEAM PERFORMANCE

In this section, we present the results of an observational study on the effect of ACIV on human-AI team performance using the dataset from Vodrahalli et al. (2022).

**Data description.**  We experiment with the publicly available Human-AI Interactions dataset (Vodrahalli et al., 2022). The dataset comprises 34,783 unique predictions from 1,088 different human participants on four different binary prediction tasks ("Art", "Sarcasm", "Cities" and "Census"). In each of the tasks, human participants provide confidence values about their predictions before ($d^H$) and after ($d^{H+AI}$) receiving AI advice from a classifier in the form of the classifier's confidence values ($d^{AI}$). $d^H$, $d^{H+AI}$ and $d^{AI}$ are in the range of $[0, 1]$.

**Results.** We present the ACIV of the AI confidence values ($d^{AI}$) over human decisions ($d^H$) measured by the payoff function as the mean squared error (MSE) in Figure 4, and show the reduction of the mean squared error (MSE) of the human-AI team over the human-alone baseline in Table 2. We find that between these four tasks, the ACIV of the AI over human decisions predicts the improvement of the human-AI team performance over the human-alone baseline.

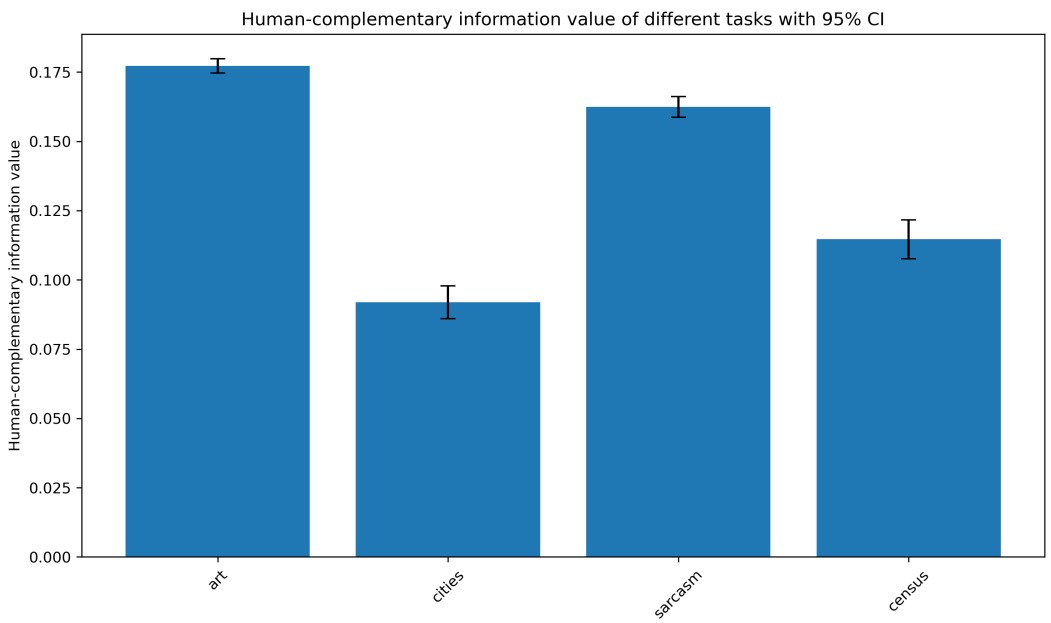

Figure 4: The ACIV of the AI confidence values over human decisions for different tasks. The error bars are 95% confidence intervals.

Table 2: The ACIV of the AI confidence values over human decisions and the reduction of mean squared error (MSE) of the human-AI team over the human-alone baseline for different tasks.

| Task | Human-complementary Info | Human MSE | Human+AI MSE | Reduction in MSE |
|---|---|---|---|---|
| Art | 0.1772 [0.1747, 0.1798] | 0.243 | 0.1847 | 0.0583 |
| Cities | 0.0919 [0.0860, 0.0978] | 0.1955 | 0.1615 | 0.034 |
| Sarcasm | 0.1625 [0.1587, 0.1662] | 0.2137 | 0.1767 | 0.037 |
| Census | 0.1147 [0.1077, 0.1217] | 0.1997 | 0.1861 | 0.0136 |

# E   STATISTICAL TESTS ON EXPERIMENT RESULTS

We calculate the p-values of the significance tests with Welch's t-test, with $\alpha = 0.05$. The degrees of freedom ($\nu$) of Welch's t-test is calculated with the following formula:

$$\nu = \frac{(\frac{s_1^2}{N_1} + \frac{s_2^2}{N_2})^2}{\frac{s_1^2}{N_1^2(N_1-1)} + \frac{s_2^2}{N_2^2(N_2-1)}} \tag{10}$$

where $s_1$ and $s_2$ are the standard deviations of the two samples, and $N_1$ and $N_2$ are the sizes of the two samples (see sample sizes and standard deviations in Table 5). We used Bonferroni correction as the multiple comparisons correction to control the overall type I error rate. The results are shown in Table 3 and Table 4.

Table 3: 95% confidence intervals for the reduction of APE with different AI models and explanations in Figure 2.

|  | No explanation | SHAP | ILIV-SHAP + SHAP |
|---|---|---|---|
| AI1 | 5.96% [5.50%, 6.42%] | 5.88% [5.47%, 6.28%] | 6.94% [6.50%, 7.38%] |
| AI2 | 6.33% [5.90%, 6.76%] | 6.22% [5.71%, 6.72%] | 5.31% [4.80%, 5.83%] |

Table 4: P-values for significance tests of reduction in APE between experimental conditions (after Bonferroni correction).

| Null Hypothesis ($H_0$) | P-value |
|---|---|
| AI1 + ILIV-SHAP + SHAP < AI1 + SHAP | 0.002 |
| AI1 + ILIV-SHAP + SHAP < AI1 + No explanation | 0.010 |
| AI1 < AI2 | 0.555 |
| AI1 + ILIV-SHAP + SHAP < AI2 + ILIV-SHAP + SHAP | <0.001 |

## F  COGNITIVE LOADS AND ANCHORING EFFECTS IN EXPERIMENT

We also examine the spent time on task and the degree of anchoring on the human versus AI decisions by looking at the distance between human-AI team decisions and AI or human-alone decisions.

**Time spent.**  Table 6 shows that there is no significant increase in the duration of the experiment for ILIV-SHAP (23.3 [13.1, 32.1] minutes for AI1 with ILIV-SHAP and 22.2 [11.9, 28.0] minutes for AI2 with ILIV-SHAP, where the square brackets show the 25% and 75% quantiles).

**Anchoring effects.**  Table 6 also shows that ILIV-SHAP does not increase anchoring on the AI or human alone decisions, while the presence of SHAP alone tends to make the participants anchor more on the AI model. With no explanation, the participants anchor more on their own decisions in the first round.

## G  DEMONSTRATION I: MODEL COMPARISON ON CHEST RADIOGRAPH DIAGNOSIS

We apply our framework to a well-known cardiac dysfunction diagnosis task (Rajpurkar et al., 2018; Tang et al., 2020; Shreekumar, 2025). We demonstrate how our framework can be used in model evaluation for analyzing how much complementary information value a set of possible AI models offers to the radiology reports written by experts.

### G.1  DATA AND MODEL

We use data from the MIMIC dataset (Goldberger et al., 2000), which contains anonymized electronic health records from Beth Israel Deaconess Medical Center (BIDMC), a large teaching hospital in Boston, Massachusetts, affiliated with Harvard Medical School. Specifically, we utilize chest x-ray images and radiology reports from the MIMIC-CXR database (Johnson et al., 2019) merged with patient and visit information from the broader MIMIC-IV database (Johnson et al., 2023). The payoff-related state, cardiac dysfunction $\omega \in \{0, 1\}$, is coded based on two common tests, the NT-proBNP and the troponin, using the age-specific cutoffs from Mueller et al. (2019) and Heidenreich et al. (2022). We label the radiology reports by a rule-based tool (Irvin et al., 2019) and use the labels as the human decisions (without AI assistance) in the diagnosis task to solve the problem of computational feasibility with high-dimensional textual reports. The labels are represented by the symptoms as positive, negative, or uncertain, i.e., $d \in \{+, ?, -\}$. We fine-tuned five deep-learning models on the cardiac dysfunction diagnosis task, VisionTransformer (Dosovitskiy, 2020), SwinTransformer (Liu

Table 5: Sample sizes and standard deviations of the reduction of APE for each experimental condition

| Model | Explanation | Participants | Observations (N) | Standard Deviation |
|---|---|---|---|---|
| AI1 | ILIV-SHAP + SHAP | 70 | 840 | 0.0653 |
| | SHAP | 79 | 948 | 0.0642 |
| | No explanation | 69 | 828 | 0.0668 |
| | Total (all explanations) | 218 | 2,616 | 0.0656 |
| AI2 | ILIV-SHAP + SHAP | 67 | 804 | 0.0746 |
| | SHAP | 62 | 744 | 0.0679 |
| | No explanation | 74 | 888 | 0.0673 |
| | Total (all explanations) | 203 | 2,436 | 0.0700 |
| Total | | 421 | 5,052 | 0.0678 |

Table 6: The number of participants (after filtering based on the criteria in the pre-registration), the mean duration of the experiment, and the anchoring on AI and human for each condition. Anchoring on AI is calculated as the mean of the absolute difference between the human-AI team's prediction and the AI's prediction normalized by the actual sale price, $|d^{\text{Human-AI}} - d^{\text{AI}}|/\omega$, and anchoring on human is calculated as the mean of the absolute difference between the human-AI team's prediction and the human's prediction normalized by the actual sale price, $|d^{\text{Human-AI}} - d^{\text{Human}}|/\omega$. The square brackets indicate the 25th and 75th percentiles.

| Condition | # of participants | Mean Duration (minutes) | Anchoring on AI ($\downarrow$) | Anchoring on Human ($\downarrow$) |
|---|---|---|---|---|
| AI1 + ILIV-SHAP+SHAP | 70 | **23.3** [13.1, **32.1**] | 0.303 [0.0277, 0.253] | 0.404 [0.0746, 0.491] |
| AI1 + SHAP | 79 | 21.3 [12.0, 27.9] | **0.211** [**0.0201**, **0.222**] | 0.427 [**0.0721**, 0.533] |
| AI1 + No Explanation | 69 | 22.4 [**14.5**, 28.4] | 0.226 [0.0276, 0.258] | **0.367** [0.0740, **0.483**] |
| AI2 + ILIV-SHAP+SHAP | 67 | 22.2 [11.9, 28.0] | 0.206 [0.0273, 0.260] | 0.458 [0.0840, 0.613] |
| AI2 + SHAP | 62 | **23.1** [**13.2**, 27.0] | **0.203** [**0.0215**, **0.224**] | 0.456 [0.0777, 0.602] |
| AI2 + No Explanation | 74 | 22.5 [12.7, **30.5**] | 0.259 [0.0265, 0.260] | **0.434** [**0.0611**, **0.501**] |

et al., 2021), ResNet (He et al., 2016), Inception-v3 (Szegedy et al., 2016), and DenseNet (Huang et al., 2017). Our training set contains 12,228 images, and the validation set contains 6,115 images. On a hold-out test set with 12,229 images, the AUC achieved by the five models is: DenseNet with 0.77, Inception v3 with 0.76, ResNet with 0.77, SwinTransformer with 0.78, and VisionTransformer with 0.80.

We consider Brier score, a.k.a., quadratic score, as the payoff function: $S(\omega, d) = 1 - (\omega - d)^2$. The scale of the quadratic score is $[0, 1]$ and a random guess ($d \sim \text{Bernoulli}(0.5)$) achieves 0.75 payoff. We use the quadratic score instead of the mean absolute error that is usually used in cardiac dysfunction diagnosis task because the quadratic score is a proper scoring rule where truthfully reporting the belief maximizes the payoff[11]. We also conduct a robust analysis considering various V-shaped payoff functions with different kinks on a discretized grid of $[0, 1]$ with a step of $0.01$. We use the hold-out test set to estimate the data-generating process, which defines the joint distribution of state, human decisions, and AI models' predictions.

We construct the scale of performances by a no-information bound and a full-information bound. The no-information bound is $\text{R}^{\pi,S}(\emptyset)$, the baseline as we define the information value. The full-information bound is defined as the expected payoff of a rational DM who has access to all signals, human label from radiology report and predictions from five AI models.

## G.2 RESULTS

**Can the AI models complement human judgment?** We first analyze the agent-complementary information values in Figure 5, using the Brier score as the payoff function. We find that all AI models provide complementary information value to the aforementioned human judgment. As shown

---

[11]We prefer a proper scoring rule so that the rational decision-maker's strategy is to reveal their true belief, ensuring that the signal's information value accurately reflects its role in forming beliefs.

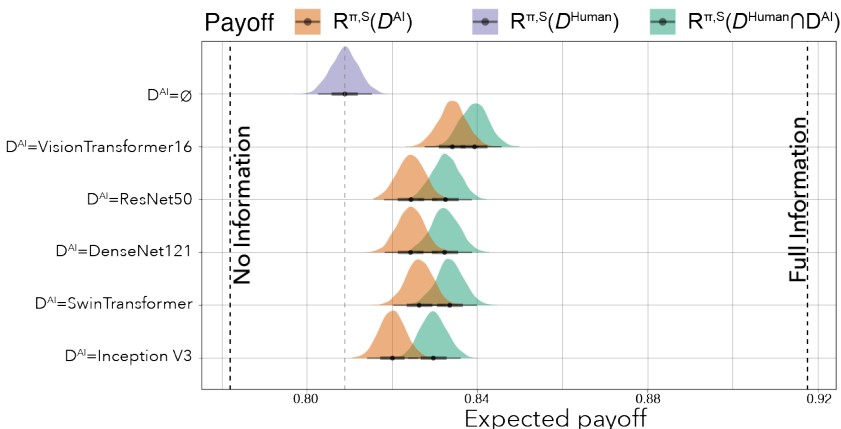

Figure 5: Information value of all deep-learning models calculated under our framework. The first row represents the human-alone decisions (without considering any AI predictions as additional signals). The other rows are the combinations of the human-alone decisions and the AI predictions from different pre-trained models. We list the AI predictions alone to show the AI-complementary information value offered by human decisions.

in Figure 5 (comparison between $\mathrm{R}^{\pi,S}(D^{\text{Human}} \cup D^{\text{AI}})$ and $\mathrm{R}^{\pi,S}(D^{\text{Human}})$), all AI models capture at least $20\%$ of the total available information value (across all AI model and human decisions) that is not exploited by human decisions. This motivates deploying an AI to assist humans in this scenario.

In the other direction, the human decisions also provide complementary information to all AI models, comparing $\mathrm{R}^{\pi,S}(D^{\text{Human}})$ with $\mathrm{R}^{\pi,S}(D^{\text{AI}})$ in Figure 5. This observation might inspire, for example, further investigation of the information the humans can access to that is not represented in AI training data.

**Which AI model offers the most decision-relevant information over human judgments?** Figure 5 shows that VisionTransformer contains slightly higher information value than the other models, and Inception v3 contains slightly lower information value than the other models. We assess the stability of VisionTransformer's superiority over the other AI models across many possible losses to test if there is a Blackwell ordering of models. By Proposition B.3, we test the payoff of models on all V-shaped scoring rules, shown in Figure 6. Across all the V-shaped payoff functions, we find that VisionTransformer is Blackwell more informative and Inception v3 is Blackwell less informative than all other models. The VisionTransformer achieves a higher information value on all V-shaped scoring rules, implying a higher information value on all decision problems.

## H    DEMONSTRATION II: BEHAVIORAL ANALYSIS ON DEEPFAKE DETECTION

We apply our framework to analyze a deepfake video detection task (Dolhansky et al., 2020), where participants are asked to judge whether a video was created by generative AI, including with the assistance of an AI model.

### H.1    DATA AND MODEL

We define the information model on the experiment data of Groh et al. (2022). Non-expert participants (n=5,524) were recruited through Prolific and asked to examine the videos. They reported their decisions in two rounds. They first reviewed the video and reported an *initial* decision ($D^{\text{Human}}$) without access to the AI model. Then, in a second round, they were provided with the recommendation ($D^{\text{AI}}$) of a multitask cascaded convolutional neural network (Zhang et al., 2016), with estimated $65\%$ accuracy on holdout data, and chose whether to change their initial decision. This produced a *final* decision ($D^{\text{Human-AI}}$). Both decisions were elicited as a percentage indicating how confident the participant was that the video was a deepfake, measured in $1\%$ increments: $d \in \{0\%, 1\%, \dots, 100\%\}$.

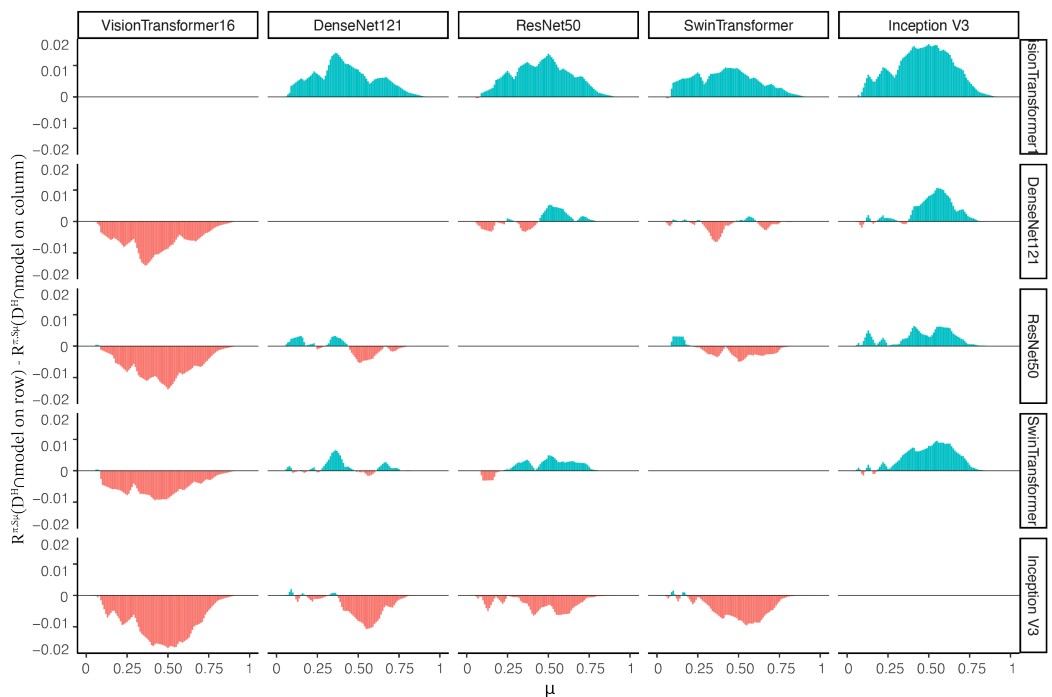

Figure 6: Robust analysis for demonstration I on all V-shaped payoff functions. The kink $\mu$ is shown on the $x$-axis. Each subplot displays the difference between the ACIV on the row model over human decisions and the ACIV on the column model over human decisions. A positive value (colored in blue) at $\mu$ indicates the model on the row contains more informative than the model on the column under the evaluation of V-shaped scoring rule with kink $\mu$. The subplots are symmetric along the diagonal, e.g., $(1, 2)$ subplot and $(2, 1)$ subplot display the same distribution with opposite signs.

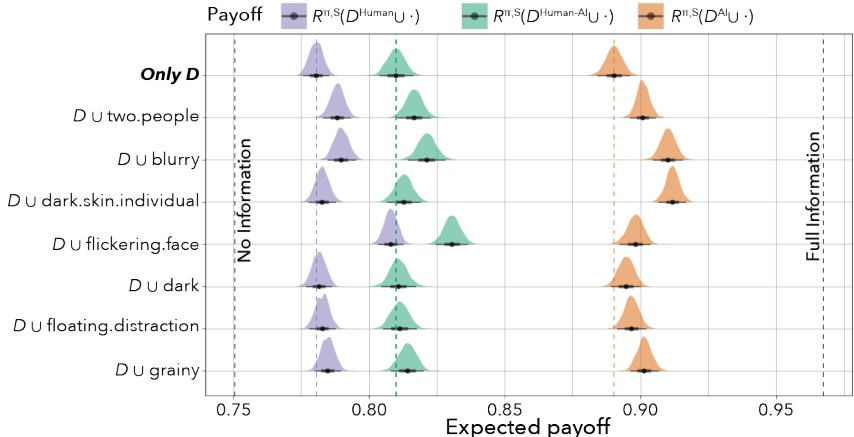

Figure 7: Information value calculated under our framework in the information model defined by the experiment of Groh et al. (2022). Basic signals include the seven video level features and three types of agent decisions. The baseline on the left represents the expected payoff given no information, i.e., $R^{\pi,S}(\emptyset)$, and the benchmark on the right represents the expected payoff given all available information, i.e., $R^{\pi,S}(\{\Sigma_1, \ldots, \Sigma_n, D^{\text{Human}}, D^{\text{Human-AI}}, D^{\text{AI}}\})$. All the payoffs are calculated by $R^{\pi,S}(\cdot)$, where $\cdot$ is the signal on the $y$-axis.

We round the predictions from the AI model to the same 100-scale probability scale available to study participants.

We use the Brier score as the payoff function, with the binary payoff-related state: $\omega \in \{0, 1\} = \{\text{genuine}, \text{fake}\}$. This choice differs from the mean absolute error used by Groh et al. (2022), but again we use the quadratic score because it is a proper scoring rule where truthfully reporting the belief maximizes the score.

We identify a set of features that were implicitly available to all three agents (human, AI, and human-AI). Because the video signal is high dimensional, we make use of seven video-level features using manually coded labels by Groh et al. (2022): graininess, blurriness, darkness, presence of a flickering face, presence of two people, presence of a floating distraction, and the presence of an individual with dark skin, all of which are labeled as binary indictors. These are the basic signals in our framework. We estimate the data-generating process $\pi$ using the realizations of signals, state, first-round human decisions, AI predictions, and second-round human-AI decisions. The no-information bound is the same as Appendix G while the full-information bound is defined as the expected payoff of a rational DM who has access to all video-level features and three agents' decisions.

## H.2 RESULTS

**How much decision-relevant information do agents' decisions offer?**   We first compare the information value of the AI predictions to that of the human decisions in the first round (without AI assistance). Figure 7(a) shows that **AI predictions** provide about $65\%$ of the total possible information value over the no-information baseline, while **human decisions** only provide about $15\%$. Because the no information baseline, $0.75$, is equivalent to a random guess drawn from Bernoulli$(0.5)$, human decisions are only weakly informative for the problem.

We next consider the **human-AI decisions**. Given that the AI predictions contain a significant portion of the total possible information value, we might hope that when participants have access to the AI predictions, their performance will be close to the full information baseline. However, the information value of the **human-AI decisions** only achieves a small proportion of the total possible information value ($30\%$). This is consistent with the findings of Guo et al. (2024) that humans are bad at distinguishing when AI predictions are correct.

**How much additional decision-relevant information do the available features offer over agents' decisions?**   To understand what information might improve human decisions, we assess the ACIVs of different signals over different agents. This describes the additional information value in the signal after conditioning on the existing information in the agents' decisions. As shown on the fifth row in Figure 7, the presence of a flickering face offers larger ACIV over human decisions than over AI predictions, meaning that human decisions could improve by a greater amount if they were to incorporate this information. Meanwhile, as shown on the fourth row in Figure 7, the presence of an individual with dark skin offers larger ACIV over AI predictions than over human decisions, suggesting that humans make greater use of this information. This suggests that the AI and human rely on differing information to make their initial predictions, where the AI relies more on information associated with the presence of a flickering face while human participants rely more on information associated with the presence of an individual with dark skin.

By comparing the ACIVs of different signals over human decisions and human-AI decisions, we also find that simply displaying AI predictions to humans did not lead to the AI-assisted humans exploiting the observed signals in their decisions. As shown in Figure 7, with the assistance of AI, the ACIVs of all signals over the human-AI teams' decisions do not change much compared to the ACIVs over human decisions, with the exception of a slight improvement in the presence of a flickering face. This finding further confirms the hypothesis that humans are simply relying on AI predictions without processing the information contained in them.

## I   SENSITIVITY ANALYSIS OF THE RATIONAL BELIEF ESTIMATOR

In this section, we first present a theoretical analysis on the quality of the ACIV estimator by connecting the calibration error of the given $\mathcal{A}$. Then, we present an empirical analysis on the chest X-ray diagnosis demonstration with different algorithms (linear regression (LR), gradient boosting methods (GBM), and neural network (NN)) as $\mathcal{A}$.

## I.1 THEORETICAL ANALYSIS

In this section, we theoretically show how the quality of the ACIV estimator (Algorithm 1) is related to the calibration error of the rational belief estimator $a = \mathcal{A}(\{v_i, d_i^b, \omega_i\}_{i=1}^n)$. We use the regret of the decision maker to represent the quality of the estimation on the rational decision rules, i.e., how much payoff can be improved if we correct the decisions in hindsight[12]. Because the ACIV is defined as the improvement of the best-attainable performance with the signal and the agent decision over the one with the agent decision alone, the decisions $\hat{d}^r$ and $\hat{d}^{rb}$ in Algorithm 1 should have as small regret as possible. We first derive our definition of the regret of Algorithm 1 using the swap regret by Roth & Shi (2024).

**Definition I.1** (Swap Regret (Roth & Shi, 2024)). Given a set of observations $\{(v_i, d_i^b, \omega_i)\}_{i=1}^n$, a decision rule $d(\cdot)$, and a decision task with payoff function $S$, the swap regret of the DM is:

$$\text{SWAP}_S(d, \{(v_i, d_i^b, \omega_i)\}_{i=1}^n) = \frac{1}{n} \max_{\sigma:\mathbf{D}\to\mathbf{D}} \sum_{i=1}^n \left[ S(\sigma(d(v_i, d_i^b)), \omega_i) - S(d(v_i, d_i^b), \omega_i) \right] \quad (11)$$

The swap function $\sigma$ is a permutation of the action space $\mathbf{D}$ that maps the action of $d^r(v_i, d_i^b)$ to another action. We define the regret of the ACIV estimator as the difference between the ACIV under the best-responding decision rules—$d^{*r}$ and $d^{*rb}$—and the ACIV under the estimated decision rules—$\hat{d}^r$ and $\hat{d}^{rb}$.

**Definition I.2** (Regret of the ACIV estimator). Given an empirical distribution as the data-generating process, $\pi = \text{Uniform}(\{v_i, d_i^b, \omega_i\}_{i=1}^n)$, the estimated decision rules $\hat{d}^r : \mathbf{V} \times \mathbf{D} \to \mathbf{D}$ and $\hat{d}^{rb} : \mathbf{D} \to \mathbf{D}$ from Algorithm 1, and a decision task with payoff function $S$, the regret of the ACIV estimator is:

$$\text{REGACIV}_{S,\pi}(\hat{d}^r, \hat{d}^{rb}; V, D^b) = \text{ACIV}^{\pi,S,d^{*r},d^{*rb}}(V; D^b) - \text{ACIV}^{\pi,S,\hat{d}^r,\hat{d}^{rb}}(V; D^b) \quad (12)$$

where $\text{ACIV}^{\pi,S,\hat{d}^r,\hat{d}^{rb}}$ denotes the ACIV estimated under the rational decision rules $\hat{d}^r$ and $\hat{d}^{rb}$. $d^{*r}$ and $d^{*rb}$ are the optimal decision rules that we can get from $\hat{d}^r$ and $\hat{d}^{rb}$:

$$d^{*r}(\cdot, \cdot) = \sigma^r(\hat{d}^r(\cdot, \cdot)) \text{ where } \sigma^r = \arg\max_{\sigma:\mathbf{D}\to\mathbf{D}} \sum_{i=1}^n \left[ S(\sigma(\hat{d}^r(v_i, d_i^b)), \omega_i) \right]$$

$$d^{*rb}(\cdot) = \sigma^{rb}(\hat{d}^{rb}(\cdot)) \text{ where } \sigma^{rb} = \arg\max_{\sigma:\mathbf{D}\to\mathbf{D}} \sum_{i=1}^n \left[ S(\sigma(\hat{d}^{rb}(d_i^b)), \omega_i) \right]$$

**Lemma I.3.** *The regret of the ACIV estimator is upper bounded by the swap regret of $\hat{d}^r$:*

$$\text{REGACIV}_{S,\pi}(\hat{d}^r, \hat{d}^{rb}; V, D^b) \le \text{SWAP}_S(\hat{d}^r, \{(v_i, d_i^b, \omega_i)\}_{i=1}^n) \quad (13)$$

*Proof.*

$$\begin{aligned}
\text{REGACIV}_{S,\pi}(\hat{d}^r, \hat{d}^{rb}; V, D^b) &= \mathbf{E}_{(v,d^b,\omega)\sim\pi}[S(\sigma^r(\hat{d}^r(v, d^b)), \omega_i)] - \mathbf{E}_{(v,d^b,\omega)\sim\pi}[S(\hat{d}^r(v, d^b), \omega_i)] - \\
&\quad \left( \mathbf{E}_{(d^b,\omega)\sim\pi}[S(\sigma^{rb}(\hat{d}^r(d_i^b)), \omega_i)] - \mathbf{E}_{(d^b,\omega)\sim\pi}[S(\hat{d}^r(d_i^b), \omega_i)] \right) \\
&\le \mathbf{E}_{(v,d^b,\omega)\sim\pi}[S(\sigma^r(\hat{d}^r(v, d^b)), \omega_i)] - \mathbf{E}_{(v,d^b,\omega)\sim\pi}[S(\hat{d}^r(v, d^b), \omega_i)] \\
&= \max_{\sigma:\mathbf{D}\to\mathbf{D}} \left[ \frac{1}{n} \sum_{i=1}^n \left[ S(\sigma(\hat{d}^r(v_i, d_i^b)), \omega_i) - S(\hat{d}^r(v_i, d_i^b), \omega_i) \right] \right] \\
&= \text{SWAP}_S(\hat{d}^r, \{(v_i, d_i^b, \omega_i)\}_{i=1}^n)
\end{aligned}$$

$$(14)$$

$\square$

---

[12]We use the notion of regret to quantify the quality of our ACIV estimator because the perfect decisions are usually unidentifiable with finite signals.

**Claim I.4** (Kleinberg et al. (2023), Theorem 12). Given a decision task with payoff function $S : \mathbf{D} \times \mathbf{\Omega} \to [0, 1]$, if the DM responds by taking $d(v, d^b) = d^*(a(v, d^b))$, where $a$ is an estimator of the probability of the payoff state given the signal and action and $d^*(p) = \arg\max_{d \in \mathbf{D}} \mathbf{E}_{\omega \sim p}[S(d, \omega)]$ is the best response to the probability $p$, the swap regret of the DM is bounded by the expected calibration error (ECE) of the estimator $a$:

$$\text{SWAP}_S(\hat{d}^r, \{(v_i, d_i^b, \omega_i)\}_{i=1}^n) \leq 2\text{ECE}(a, \{(v_i, d_i^b, \omega_i)\}_{i=1}^n) \tag{15}$$

**Theorem I.5.** *Given a decision task with bounded payoff $S : \mathbf{D} \times \mathbf{\Omega} \to [M_1, M_2]$, the regret of the ACIV estimator is bounded by the expected calibration error of the estimator $\hat{a} = \mathcal{A}(\{v_i, d_i^b, \omega_i\}_{i=1}^n)$:*

$$\text{REGACIV}_{S,\pi}(\hat{d}^r, \hat{d}^{rb}; V, D^b) \leq 2(M_2 - M_1)\text{ECE}(\hat{a}, \{(v_i, d_i^b, \omega_i)\}_{i=1}^n) \tag{16}$$

The proof normalizes the payoff in Lemma I.3 into $[0, 1]$ and then applies Claim I.4.

Theorem I.5 shows that, in Algorithm 1, when we choose a predictive algorithm $\mathcal{A}$ that yields low ECE, the ACIV estimated by Algorithm 1 is close to the ACIV under the optimal decision rules.

## I.2 EMPIRICAL ANALYSIS

In this section, we take the chest X-ray diagnosis task in Appendix G as an example to empirically analyze the estimation of the full information value by Algorithm 1 with different modeling approaches (linear regression (LR), gradient boosting methods (GBM), and neural network (NN)).

**Task and data.** We use the chest X-ray diagnosis task in Appendix G as the example. The task is to predict the probability of the presence of a disease given a chest X-ray image. The decision space is $\mathbf{D} = \Delta[0, 1]$ and the payoff space is $\mathbf{\Omega} = \{0, 1\}$. The payoff function is $S(d, \omega) = 1 - (d - \omega)^2$. The signal space is a high-dimensional feature vector of the chest X-ray image. We split the dataset that we used in Appendix G into a 70/30 train/test split.

**Predictive Models.** We use the following models:

- Linear Regression (LR) from `scikit-learn` package.
- Extreme Gradient Boosting (GBM) from `xgboost` package with `n_estimators = 100`, `max_depth = 6`, and `learning_rate = 0.3`.
- MLP classifier (NN) from `scikit-learn` package with 2 hidden layers of 256 and 64 neurons each.

**Evaluation Metrics.** We report the following evaluation metrics for each estimator:

- Brier Score: $\frac{1}{n} \sum_{i=1}^n (\hat{p}_i - \omega_i)^2$
- Accuracy: $\frac{1}{n} \sum_{i=1}^n \mathbb{1}(\mathbb{1}(\hat{p}_i \geq 0.5) = \omega_i)$
- Expected Calibration Error (ECE): $\frac{1}{n} \sum_{i=1}^n |\hat{p}_i - \mathbf{E}[\omega|\hat{p}_i]|$

Table 7: Performance of different algorithms for ACIV estimation in the chest X-ray diagnosis task. Metrics for the estimators include Brier Score, Accuracy, F1-Score, and Expected Calibration Error (ECE). We report the estimated $\text{ACIV}(V; \emptyset)$ for each estimator with Algorithm 1.

| Model | Brier Score ↑ | Accuracy ↑ | ECE ↓ | ACIV(V; ∅) ↑ |
|---|---|---|---|---|
| Linear Regression (LR) | 0.834 | 0.775 | 0.039 | 0.086 |
| Gradient Boosting (GBM) | 0.841 | 0.771 | **0.026** | **0.108** |
| Neural Network (NN) | **0.853** | **0.789** | 0.036 | 0.0757 |

**Results.** The results are shown in Table 7. We can see that the GBM estimator achieves the best performance in terms of ECE, while the NN estimator achieves the best performance in terms of Brier Score and Accuracy. This validates our theoretical result: ECE is a good proxy for the regret of the estimation of the ACIV by Algorithm 1 compared to the Brier Score and Accuracy.

## J    SCREENSHOTS OF THE EXPERIMENT

**Trial number: 11/12, Round: 1/2**

| Year Built | 2004 |
|---|---|
| # of Fireplaces | 0 |
| Car Capacity in Garage | 2.0 |
| Above Grade Living Area | 2084 sq ft |
| Feature X | 420 |
| Feature Y | 140 |

What is your guess for the sale price of this house?
(Please use the following slider to choose the answer and then
click on the Next button to continue.)

| 13 | 87 | 161 | 236 | 310 | 384 | 458 | 532 | 607 | 681 | 755 |

Price ($K)

Figure 8: Screenshot of the human-alone trials in the first round of the experiment.

**Trial number: 1/12, Round: 2/2**

| Year Built | 1920 |
|---|---|
| # of Fireplaces | 0 |
| Car Capacity in Garage | 2.0 |
| Above Grade Living Area | 1196 sq ft |
| Feature X | 500 |
| Feature Y | 120 |

**Model prediction: $144.1K**
**Your previous guess: $160K**

What is your guess for the sale price of this house?
(Please use the following slider to choose the answer and then click on the Next button to continue.)

| 13 | 87 | 161 | 236 | 310 | 384 | 458 | 532 | 607 | 681 | 755 |

Price ($K)

Figure 9: Screenshot of the no explanation condition in the second round of the experiment.

**Trial number: 1/12, Round: 2/2**

| Feature name | Value | Impact on AI prediction |
|---|---|---|
| Year Built | 2007 | +$7.7K |
| # of Fireplaces | 1 | -$7.2K |
| Car Capacity in Garage | 3.0 | +$17.6K |
| Above Grade Living Area | 1542 sq ft | -$0.1K |
| Feature X | 360 | +$7K |
| Feature Y | 160 | +$31.7K |

**Model prediction: $286.1K**
**Your previous guess: $160K**

What is your guess for the sale price of this house?
(Please use the following slider to choose the answer and then click on the Next button to continue.)

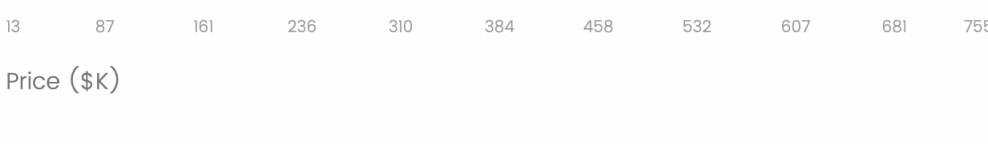

Figure 10: Screenshot of the SHAP condition of AI1 in the second round of the experiment.

**Trial number: 1/12, Round: 2/2**

| Feature name | Value | Impact on AI prediction |
|---|---|---|
| Year Built | 1940 | -$17.5K |
| # of Fireplaces | 2 | +$10K |
| Car Capacity in Garage | 2.0 | -$0.9K |
| Above Grade Living Area | 2480 sq ft | +$35.2K |
| Feature X | 1500 | $0K |
| Feature Y | 140 | $0K |

**Model prediction: $238.7K**
**Your previous guess: $160K**

What is your guess for the sale price of this house?
(Please use the following slider to choose the answer and then click on the Next button to continue.)

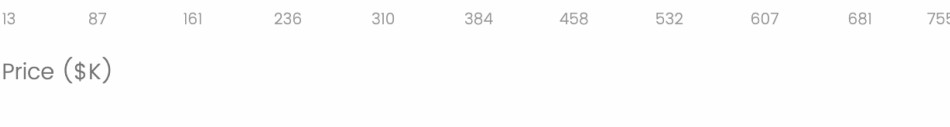

Price ($K)

Figure 11: Screenshot of the SHAP condition of AI2 in the second round of the experiment.

## Trial number: 2/12, Round: 2/2

| Feature name | Value | Impact on AI prediction |
|---|:---:|:---:|
| Feature Y | 100 | -$6.2K |
| # of Fireplaces | 0 | -$13.3K |
| Year Built | 1900 | -$7.5K |
| Car Capacity in Garage | 2.0 | -$1.1K |
| Feature X | 380 | +$4.8K |
| Above Grade Living Area | 1627sq ft | +$1K |

**Model prediction: $134.9K**
**Your previous guess: $160K**
Highlighted features are where the model gets significant amount of human-complementary information.

What is your guess for the sale price of this house?
(Please use the following slider to choose the answer and then click on the Next button to continue.)

| 13 | 87 | 161 | 236 | 310 | 384 | 458 | 532 | 607 | 681 | 755 |
|---|---|---|---|---|---|---|---|---|---|---|

Price ($K)

Figure 12: Screenshot of the ILIV-SHAP + SHAP condition of AI1 in the second round of the experiment.

**Trial number: 1/12, Round: 2/2**

| Feature name | Value | Impact on AI prediction |
|---|---|---|
| Year Built | 1941 | -$12.7K |
| Above Grade Living Area | 1376sq ft | -$2.1K |
| Car Capacity in Garage | 1.0 | -$3.7K |
| # of Fireplaces | 1 | -$3.1K |
| Feature X | 1500 | $0K |
| Feature Y | 100 | $0K |

**Model prediction: $135.3K**
**Your previous guess: $160K**
Highlighted features are where the model gets significant amount of human-complementary information.

What is your guess for the sale price of this house?
(Please use the following slider to choose the answer and then click on the Next button to continue.)

| 13 | 87 | 161 | 236 | 310 | 384 | 458 | 532 | 607 | 681 | 755 |

Price ($K)

Figure 13: Screenshot of the ILIV-SHAP + SHAP condition of AI2 in the second round of the experiment.

