# OpenReview forum: "The Value of Information in Human-AI Decision-making"
_ICLR.cc/2026/Conference — ICLR 2026 Poster_

### Official Review · Reviewer_A4pM · 2025-10-27

**Soundness:** 2
**Presentation:** 2
**Contribution:** 3
**Rating:** 4
**Confidence:** 3

**Summary:**

The paper proposes a decision-theoretic framework to quantify the *value of information* in human-AI decision workflows. It formalizes (i) *Agent-Complementary Information Value (ACIV)* (the marginal payoff gain from adding a signal (V) beyond what is already encoded in an agent's decisions ($D_b$)) and (ii) an *instance-level* analogue (*ILIV*) to analyze specific realizations. Building on ILIV, the authors introduce *ILIV-SHAP*, an explanation method that attributes feature importance to *complementary* information rather than to predictive contribution alone. A preregistered Prolific study on *house-price prediction* ($n=421$) crosses two models with different ex-ante ACIV (AI1 high vs. AI2 low) and three explanation conditions (ILIV-SHAP+SHAP / SHAP / None). Results indicate that (a) the higher-ACIV model yields larger error reductions for human-AI teams, and (b) *ILIV-SHAP+SHAP* outperforms *SHAP* alone in reducing absolute percentage error (APE). The framework is further demonstrated on *chest X-ray diagnosis* and *deepfake detection* to show how ACIV/ILIV expose complementarity patterns between humans and models.

**Strengths:**

* *Clear formalization of 'information value' in teaming.* The ACIV/ILIV definitions make the complementarity question precise, grounding it in Bayesian decision theory and proper scoring rules (incl. a robustness view). This provides a principled upper-bound benchmark and a way to diagnose under-use of available signals.
* *Practical algorithms and instance-level lens.* Algorithms 1-2 give workable estimators for ACIV/ILIV, enabling empirical analysis without requiring full generative models of all signals.
* *Explanation innovation (ILIV-SHAP).* Reframing attribution around *decision-relevant payoff gains* (rather than prediction deltas) is original and well-motivated; the method preserves SHAP axioms while targeting complementarity.
* *Empirical evidence for design claims.* In the crowdsourced study, the high-ACIV model (AI1) shows bigger APE reductions than the low-ACIV model (AI2), and ILIV-SHAP+SHAP yields larger gains than SHAP alone for AI1 (e.g., 7.11% vs. 5.20% for SHAP; CI reported). The analysis includes hierarchical modeling and multiple-comparison controls.
* *External demonstrations.* The chest-X-ray and deepfake case studies illustrate how the framework can compare agents and reveal which features/signals are under- or over-utilized by humans vs. models.

**Weaknesses:**

1. *Human-AI complementarity definition*. In the related work section, the *human-AI complementarity* term is, in my opinion, not properly defined and there is also lack of related and important related work missing [1-5]. In this area, complementarity refers to team decisions that, in expectation, outperform both the human and the AI alone [Bansal et al 2021a(already cited),1,2], rather than just analyzing the *information value and use*, as author's state in the related work. However, imo, authors' use a better definition in the first paragraph of the introduction.
2. *Complementarity by design related work*. While I understand the authors' analysis of the value of additional information from one of the agents, I believe they also make a claim in the paper that existing complementarity approaches "do not account for the best achievable performance given the information available at the time of the decision" (l. 41). I find this to be an overstatement and a shortcoming in the related work section. There are work in the literature proposing existing complementarity approaches **by design**. These methods exploit information asymmetry (where humans possess additional contextual knowledge, which the authors identify as a central aspect of human-AI complementarity) and offer principled methods with **provable guarantees** in decision-making about the better perfomance of the teaming. Specific applications of these methods include classification [3,4] or matching problems [5]. These methods effectively identify parameters that model collaboration, achieving what the authors define as the "best achievable performance given the information available at the time of the decision". I recommend revising the Related Work section to accurately position your contribution in relation to this body of research, highlighting the differences in goals, assumptions, and guarantees. Anticipating author's comments, I want to emphasize that I do not see these approaches as substitutes for the authors' contribution. Rather, *my concern is with how the introduction and related work is positioned*: some of the current statements underrepresent what prior work already achieves in this space. In fact, the authors' proposed framework (which focuses on quantifying and analyzing the value of additional information) could even complement and strengthen these existing methods. I appreciate that the authors are not proposing a new method per se but a general framework, and I believe clarifying this positioning would make the paper's contribution more precise and impactful.
3. *Estimator dependence and potential leakage.* ACIV/ILIV are estimated via learned regressors ($\hat a, \hat a_b$). The main text does not detail safeguards (strict train/validation/test splits aligned with the team-vs-human comparisons, calibration of posteriors, or sensitivity to the choice of (A)). Since the conclusions rely on these regressors approximating posteriors used in the rational benchmark, a *sensitivity analysis* (alternate model classes, calibration checks, and ablations on (A)) would strengthen validity.
4. *Flow of theory section.* The notation and explanation is dense and introduces granularly: mixing state, signals, decisions, and payoffs  are presented and used before an anchoring example. The exposition would benefit from a *running toy example* (e.g., the umbrella example) threaded through Defs. 3.1-3.3 and Algorithms 1-2, plus a *table of symbols* and an 'intuitions first' paragraph for ACIV vs. ILIV.
5. *SHAP computation*. There is no optimization on the SHAP computation: shap is O(2^n) in the number of features, so for high-dimensional data this can be very slow. Some approximation methods exist (e.g., sampling-based), but they are not discussed. In addition, the paper does not address the potential for bias in the SHAP values (e.g. not stable to noise) [6], which could impact the fairness of the model.
6. *Intervention realism in the main user study.* The two 'uninterpretable' features (Feature X/Y) are relabeled to be intentionally opaque, which creates complementarity by construction. This is a clean testbed but may *overstate* ILIV-SHAP benefits relative to realistic domains where human observables partially correlate with those features. I'd appreciate if authors discuss this and add a variant where 'semi-interpretable' proxies are available to humans.
7. *Cognitive load and UI details for ILIV-SHAP.* The paper states that ILIV-SHAP 'ranked and highlighted' features exceeding a threshold, but there's limited information on *interface design*, thresholds, and how users processed both SHAP and ILIV-SHAP simultaneously. Add representative screenshots of the UI and an *attention/time* analysis to verify that added information did not induce confusion or anchoring.
8. *Ethical nuance in DeepFake study.* In the deepfake study, the discussion of the 'dark skin' feature as informative should more explicitly acknowledge fairness and representation concerns and potential spurious correlations when translating these insights into deployed systems. Even if out of scope, a short cautionary note would be responsible. Authors could add a brief fairness statement in the deepfake section.

*Minor*:

* Some key p-values and sample sizes appear only in Appendix D/Table 2-3. Bring the primary contrasts (AI1 vs. AI2; ILIV-SHAP+SHAP vs. SHAP) with effect sizes and corrected p-values into the main text to improve readability and soundness of the results.
* Fig 1 left and Fig 2 are a bit hard to read due to small fonts.
* Bring key *CIs/p-values* and *N per condition* from Appendix D into the main text/figure caption.
* ACIV abbreviation is mentioned in the abstract before being defined.
* Missing space in line 135 "...available to a DM, including..."

In sum, the paper tackles an important and timely problem (measuring complementarity in human-AI teams) with a principled framework (ACIV/ILIV) and a neat explanatory tool (ILIV-SHAP). The conceptual contribution is solid and potentially influential. The empirical study is well-designed and aligns with the theory, and the demonstrations are informative. I'm slightly below the bar primarily due to (i) limited sensitivity analyses for the learned estimators behind ACIV/ILIV, (ii) the constructed complementarity in the user study, (iii) presentation density in the theory sections, and (iv) concerns in the related work and introduction. Addressing the listed weaknesses would likely move this into acceptance.

> [1] Bayesian modeling of human-AI complementarity. PNAS 2022.

> [2] Complementarity in human-AI collaboration: Concept, sources, and evidence. EJIS 2025.

> [3] Improving Expert Predictions with Conformal Prediction. ICML 2023.

> [4] Towards human-AI complementarity with prediction sets. NeurIPS 2024.

> [5] Towards Human-AI Complementarity in Matching Tasks. HLDM @ ECML-PKDD 2025.

> [6] On the failings of Shapley values for explainability. IJAR 2024.

**Questions:**

1.  How sensitive are ACIV/ILIV estimates to the choice of model class and calibration? Could you report a *model-sweep* (e.g., LR vs. GBM vs. NN) and reliability diagrams for posterior quality used in Eq. (1)/(2)?
2.  What exact *thresholding* produced the highlighted features, and how were ties/instability handled across permutations in the SHAP-style computation? Please include a *UI figure* and an ablation where only ILIV-driven highlights (without SHAP bars) are shown.
3. Did you measure *time-on-task*, confidence, or reliance behaviors to ensure gains weren't due to simple anchoring on model outputs? Any heterogeneity across participants (e.g., numeracy)?
4. Can you replicate the user-study result on a setting where human-visible proxies to X/Y exist (weaker complementarity), and test whether ILIV-SHAP still outperforms SHAP?

---

> ### Author Response · Authors · 2025-11-18
>
> Thank you for your thoughtful comments and constructive questions. We revised our paper based on your feedback (marked in blue). We will respond to them in order of the listed weaknesses.
>
> Human-AI complementarity definition and complementarity by design. We agree that complementarity is best defined as performance of the human-AI team that exceeds either agent in isolation, and will make sure this is clearly defined upon first usage of the term in the paper. We also agree that our explication of how prior work has approached complementary by design could be better, and would help clarify what’s novel in our work. Our work is distinguished from many other approaches to achieving human-AI complementarity in that rather than focusing purely on producing complementary performance, we quantify complementary information in a decision theoretic sense, and show how intervening on complementary information can lead to improved decisions. At a high level, our definition identifies and quantifies information asymmetries, which are a common motivation in human-AI collaboration but are often left uncharacterized. The framework is flexible and can be used to audit complementary information before deploying the human-AI collaboration systems and to check how much potential remaining complementary information stands to be exploited in theory after collecting human-AI team decisions, as we demonstrate in section 6.1 and 6.2 respectively. We revised the introduction and related works to better distinguish our work from others and acknowledge the examples the reviewer suggests.
>
> Estimator dependence and potential leakage. Our work provides a general framework, with Algorithm 1-2 proposed as the estimator of the rational agent belief for high dimensional signals like images or text. We didn’t include the discussion for these estimators because of page limitations, which we acknowledge was an oversight. The main property we care about in the estimator is calibration, as the rational agent uses that prediction as Bayesian posterior probability. We revised the paper to add this (see the added appendix H). The previous draft doesn’t use the rational belief estimator in the experiment and demonstrations, but as we describe above in our response to R1, we have revised the full information benchmark for the chest x-ray task by using Algorithm 1. Here, in appendix H, we also added a comparison result between LR vs. GBM vs. NN for it with calibration error and the rational agent performance under the estimated belief (Question 1).
>
> Flow of theory section. We revised section 3 by adding a running toy example on the umbrella example–the changes are marked in blue. Thank you for the suggestion, we agree it will help.
>
> SHAP computation. We revised the paper to discuss SHAP approximation methods in general (l. 349-353). Any model-agnostic method can be applied to ILIV-SHAP. In the experiment, we calculated the exact SHAP values since the number of features is low.  We also note that ILIV-SHAP has better stability across permutations than SHAP (Question 2). The intuition behind this is that ILIV is expected to be non-decreasing as more features are included in $x’$.
>
> Intervention realism in the main user study. In the experiment, the features X/Y are only censored on labels and their values are changed by a deterministic function. Therefore, they remain correlated with other interpretable features available to the human. As the reviewer notes, our experiment is designed to provide complementarity by construction in order to function as a proof of concept. We agree it would be interesting to evaluate the impact of ILIV-SHAP in other scenarios, such as where the human might have additional private information that correlates with the censored variables, particularly if this could be studied in a real world deployment. We note that even though participants in our study had access to four variables that were not censored, the ILIV-SHAP explanations still sometimes highlighted these variables because participants did not use them very effectively, suggesting that even with fully interpretable feature, people can still benefit from complementary explanations.
> Cognitive load and UI details for ILIV-SHAP.
>
> It is our oversight that we neglected to provide these details. We decided the threshold based on the incentive provided to participants. We only highlight the features that can lead to a at least $0.25 boost on the bonus to participants, which is translated into 7.5*10^8 MSE as the threshold for ILIV-SHAP values (Question 2). We revised the paper to clarify this.
> We also added the results to show spent time and the anchoring effect on AI for each condition (Appendix E) (Question 3).
> There are a few screenshots of the interface for each condition in Appendix H (Question 2). We added the screenshots for ILIV-SHAP in the main text in revision.
>
> Ethical nuance in DeepFake study. We added a cautionary note to the deepfake demonstration.

---

> ### Comment · Reviewer_A4pM · 2025-11-19
>
> Thank you for the detailed rebuttal and for revising the manuscript. Many of my earlier concerns now feel resolved, minor, or acceptable for future work (definition and positioning of human-AI complementarity, related work, estimator dependence, theory exposition with the umbrella example, SHAP feasibility, basic UI transparency, and the ethical note on the deepfake study). Below, I only comment on a few issues that I see as partially addressed or best framed as limitations.
>
> **SHAP / ILIV-SHAP limitations and fairness (point 5)**.
> Your clarification on computational aspects and the observation that ILIV-SHAP is more stable across permutations are helpful. However, since ILIV-SHAP still relies on SVs, it likely inherits some of the broader limitations discussed in the literature. I'd suggest explicitly acknowledging that ILIV-SHAP does not resolve those foundational issues but rather reorients attribution toward complementary, decision-relevant information.
>
> **Realism of the intervention and constructed complementarity (point 6)**.
> The clarification that X/Y are label-censored via a deterministic transformation and remain correlated with interpretable features, and the explicit framing of the study as a PoC with complementarity by construction, are convincing. I would simply recommend being explicit that external validity is limited: the current user study demonstrates ILIV-SHAP in a structured, partially synthetic setting, and further work is needed to test it in more naturally arising complementary scenarios (e.g., with semi-interpretable proxies or real deployments).
>
> **Summary**. Overall, the rebuttal and revisions substantially improve the paper and resolve most of my major concerns. The points above primarily address the need to clearly identify remaining limitations and future directions, and I would now be comfortable with acceptance.

---

> > ### Author Response · Authors · 2025-11-20
> >
> > Thank you for reading our rebuttal and revision. We just revised the paper and acknowledged these two points.

---

### Official Review · Reviewer_r9GS · 2025-10-30

**Soundness:** 2
**Presentation:** 3
**Contribution:** 2
**Rating:** 2
**Confidence:** 4

**Summary:**

This paper offers a decision-theoretic approach to analyze (and improve) human-AI decision-making by quantify the "value of information" employed by each (AI or human) agent. It introduces a Bayesian decision-theoretic framework for evaluating the performance of agents against a "rational" benchmark that represents the best-attainable payoff given a set of signals. It also proposes new metrics based on this framework: Agent-Complementary Information Value (ACIV), to quantify the global value an (AI) agent adds to (human's) existing information, and Instance-Level ACIV (ILIV), to quantify this value for a specific decision. ACIV is formalized as an upper bound using a learned surrogate for the unknown data-generating process, while ILIV is operationalized by a counterfactual payoff definition and an estimator. The authors further propose ILIV-SHAP, a variant of popular SHAP explanation method to highlight features that contribute most to the AI's "complementary" information value, rather than just its prediction. The authors validate this framework through a crowdsourced experiment on house price prediction, claiming that ACIV can identify the better AI teammate and that ILIV-SHAP improves team performance more than standard SHAP. Demonstrations on chest X-ray and deepfake detection are also provided.

**Strengths:**

- **[S1] Novel and Important Problem:** The paper tackles a central problem in human-AI interaction: the failure to achieve complementarity.. The motivation to move beyond simple accuracy metrics and develop a formal theory of "information value" is both strong and timely.
- **[S2] Crisp formalism:** The core idea of defining a Bayesian rational benchmark, $R(V)$, as the best-attainable performance is theoretically sound and provides an elegant upper bound to quantify agent sub-optimality, as noted in prior work (Guo et al., 2024). Algorithmic recipes make the framework implementable in principle, and the proposal of ILIV-SHAP that mirrors SHAP while targeting complementarity is conceptually interesting (offering a nice shift from "why did the AI make this prediction?" to "why should the human listen to this prediction?").
- **[S3] Broader demonstrations.** The chest-X-ray and deepfake case studies help situate the framework beyond the main user study.

**Weaknesses:**

-  **[W1] Weak and Confounded Experimental Validation of ILIV-SHAP:** The primary experiment (Section 5) designed to validate ILIV-SHAP (RQ2) suffers from multiple issues, making it impossible to attribute the observed performance lift to the proposed method.
    * **Problematic Baseline:** The results for the main AI model (AI1) show that "No explanation" (6.50% error reduction) significantly *outperformed* "SHAP" alone (5.20%). This critical finding, which suggests standard SHAP was actively *harmful* to user performance, is reported in the plot but never discussed. The paper's headline claim that ILIV-SHAP + SHAP (7.11%) is superior to SHAP (5.20%) is misleading, as it uses a broken baseline. The more meaningful comparison to "No Explanation" (7.11% vs. 6.50%) shows a much smaller improvement, which is neither statistically nor practically significant.
    * The experiment does not compare "SHAP" vs. "ILIV-SHAP," and appears to run into **additive information confound** instead. It compares "SHAP" (control) vs. "ILIV-SHAP *and* SHAP" (treatment). The treatment group received strictly more information (two explanations) than the control (one). The reported improvement could simply be an effect of information dosage, not the specific content of ILIV-SHAP. The two conditions are not presented identically, leading to possible **presentational confound**. The SHAP condition is a standard table (Fig. 9). The ILIV-SHAP + SHAP condition includes *visual highlighting* of specific rows and a *direct textual hint* (Fig. 11). This simple, salient cue is a powerful intervention on its own and is not part of the core ILIV-SHAP contribution. The performance gain is likely attributable to this simple UI intervention, not a deep understanding of information value.

- **[W2] The Practical Utility of ACIV is Overstated:** The data shows that while ACIV may predict the direction of improvement, it is a poor predictor of the *magnitude*. In the main experiment (Fig 1, left), AI1 has more than double the ACIV of AI2. This massive theoretical gap in complementarity translated to a *trivial difference* in actual team performance (6.27% vs 5.65% error reduction, collapsing across explanations). The observational study in Appendix (Table 1) shows the same weak link. The Art task has the highest ACIV (0.1772) and highest MSE reduction (0.0583). However, "Sarcasm" has a similarly high ACIV (0.1625) but its MSE reduction (0.037) is much smaller, and nearly identical to that of "Cities" (0.034), which had a far lower ACIV (0.0919). This weak correlation between the *metric's magnitude* and the *outcome's magnitude* severely limits its practical value for model selection.

- **[W3] Weak "Human-AI" Framing and Reliance on an Idealistic Human Model:** The paper's "human-AI" framing is a significant weakness, as the core framework is general to any two agents (e.g., AI-AI) and fails to capture the unique and well-documented imperfections of human cognition.
    * The framework's reliance on a "rational Bayesian DM" as the benchmark for optimal performance is an idealistic setup that is known to be a poor model for real human decision-makers, who are subject to cognitive biases and do not perform perfect Bayesian updates.
    * Because the "human" component is modeled so abstractly, the paper fails to clearly position its contribution within specific, established human-AI collaboration paradigms, such as AI-assisted decision-making (where the human makes the final choice) versus learning-to-defer (where the system decides between agents) vs other "human-in-the-loop" options. Clear positioning would dictate which of the human-AI decision-making methods and baselines are relevant, and could require deeper engagement with additional literature (Noti et al., Zhang et al., Mahmood et al. etc.).
    * A more rigorous evaluation in a controlled AI-AI setting might have been more effective at validating the core information-theoretic framework, as the current human experiment introduces significant, unmodeled "human" noise that obscures the results.

- **[W4] Serious Practical and Computational Hurdles are Ignored:** The entire framework relies on a regression algorithm $\mathcal{A}$ to proxy the "rational Bayesian DM" and estimate the posterior $Pr[\omega | V]$.
    * **Estimator Sensitivity:** The accuracy of ACIV and ILIV is critically dependent on the choice and quality of this surrogate model $\mathcal{A}$. The paper provides no sensitivity analysis on this choice. If $\mathcal{A}$ is a poor estimator, the entire framework produces meaningless numbers.
    * **Intractability of ILIV-SHAP:** The ILIV-SHAP calculation (Definition 4.1) appears computationally intractable. It requires a combinatorial number of calls to the ILIV function. The ILIV function itself (Algorithm 2) requires conditioning on an exact signal match $v_j = v$. In any high-dimensional or continuous-feature domain, the probability of an exact match is zero. This means the set of matching instances $k$ will be 1 or 0, making the estimate undefined or infinitely high-variance. This is not a minor detail, and suggests the proposed method is unusable for most non-trivial problems.


[Noti et al.] "Learning When to Advise Human Decision Makers." In Proceedings of the 2023 International Joint Conference on Artificial Intelligence.

[Zhang et al.]. "Learning to Complement with Multiple Humans." arXiv preprint arXiv:2311.13172 (2023).

[Mahmood et al.] "Designing behavior-aware AI to improve the human-AI team performance in AI-assisted decision making." In Proceedings of the 2024 International Joint Conference on Artificial Intelligence.

**Questions:**

- **Theoretical Inaccuracy in Appendix:** The appendix introduces a greedy algorithm (Algorithm 3) and claims that "it orders the signals the same as the Shapley value". This appears incorrect. A greedy algorithm finds a single ordering that approximates the maximum set value, whereas the Shapley value is a unique value attribution derived from *averaging* marginal contributions over *all possible* orderings. The claim indicates a conceptual misunderstanding and should be removed/corrected.

- **Causal overreach in demonstrations.** In deepfake detection, the paper infers *why* humans underuse certain information (e.g., "simply relying on AI predictions"), but ACIV is *observational* and cannot identify causal cognitive mechanisms. Conclusions should be tempered.

---

> ### Author Response · Authors · 2025-11-18
>
> Thank you for the comments and questions. While we appreciate the feedback, there seems to be some confusion about some of our methods and claims. We will respond in order of the listed weaknesses, making clear where we do not agree with the reviewer’s reasoning.
>
> First, our experiment shows that for a model with information that could complement human judgment, ILIV-SHAP leads to greater reduction in error for house price prediction than either SHAP alone or No explanation. The review seems to argue that because ILIV-SHAP had more information than SHAP, the result does not hold. But nowhere are we arguing that ILIV-SHAP has equivalent information to SHAP, nor do we see why we would need to argue this. We show that extending traditional SHAP explanations with information about what features are most likely to complement humans (using ILIV-SHAP) leads to measurably better decisions than SHAP alone for a house price prediction task similar to that studied in prior work. We are not sure what to make of the implication that experiments should only compare treatments if they are perfectly informationally equivalent – few experiments related to designing better AI prediction interfaces would pass this criteria! We therefore strongly disagree with the reviewer’s claim that it is “impossible to attribute the observed performance lift to the proposed method”: it is absolutely possible for our randomized preregistered experiment to attribute the better decision performance with ILIV-SHAP over SHAP to the information the ILIV-SHAP condition provided about complementary information - this is why randomized controlled designs are used.
>
> The reviewer’s comment about the lack of significant difference between None and ILIV-SHAP–presumably from the results as reported in the main text–made us realize that we made a mistake in that figure. Note that the results as reported in the Appendix differed from that figure. In going back over our code to reconcile the difference, we found that we did not account for the random effect of participant id in generating the figure, but we did account for it in generating the results table in the Appendix (as is appropriate since the figure should present expected results by condition while controlling for subject-specific effects). We will update the figure in the paper. In other words, our results do show a significant improvement in performance of ILIV-SHAP over No explanation. Relative to No explanation the estimates for SHAP are lower, but not significant.

---

> ### Author Response · Authors · 2025-11-18
>
> Regarding the lack of improvement from SHAP over No explanation, prior meta-analyses of explainable AI find that many common explanations do not impact the effectiveness of human–AI collaboration [1], so the ineffectiveness of SHAP explanations alone here is not entirely surprising. Our work shows that intervening on complementary information through the explanation can offset this.
>
> The reviewer argues that “the Practical Utility of ACIV is Overstated”, because the magnitude of ACIV and the magnitude of error reduction in the demonstrations are not exactly aligned. We are not sure why one would expect ACIV to perfectly predict reduction in error, as they are two different measures. Beyond demonstrating its use for analyzing information use, we show that ACIV, a measure computed assuming theoretical best use of information, can be used to drive interventions–including model selection or explanation generation–that stand to improve decision-making.
>
> The methods we present can be applied to any setting where judgments from one or more agents inform decisions, as we mention in the discussion (last paragraph page 9). We focus in the paper discussion on the human-AI scenario because it’s a source of long-standing puzzles related to what information human experts may have access to and how it relates to an AI’s information access and use. Readers could easily apply the framework to other scenarios if we want. We do not agree that focusing on human-AI scenarios is a weakness: we think more ML-style work showing methods that can improve human-AI decision-making are warranted given the many scenarios where decisions cannot easily be automated. Our focus is most clearly aligned with work on AI-assisted decision-making (which we describe in the first paragraph of related works), where a human decision-maker has access to independent human judgments (often their own) and AI predictions. We will add a sentence making clear our framework can be applied to decision-scenarios with only artificial agents in the Introduction.
>
> The reviewer seems to misunderstand how the framework uses the rational Bayesian DM. They state that “this is an idealistic setup that is known to be a poor model for real human decision-makers, who are subject to cognitive biases and do not perform perfect Bayesian updates.” We agree that humans are not perfectly rational Bayesian decision-makers, and that the measures we compute are based on best attainable performance according to rational standards. Nowhere in the paper are we claiming that people conform to these standards. However, people do not need to conform to these standards for the benchmarks to be used productively, both to aid understanding by showing how humans deviate from a well-understood standard, and to drive measurement improvements in human decision-performance. That the latter is possible despite the difference between a rational Bayesian agent and humans attests to the value of the framework.
>
> The reviewer states that “A more rigorous evaluation in a controlled AI-AI setting might have been more effective at validating the core information-theoretic framework, as the current human experiment introduces significant, unmodeled "human" noise that obscures the results.” We again disagree that this is a weakness. The fact that we can show the value of ACIV in a setting where the decision-makers are humans (and one of the sources of predictions is also humans) is better considered a strength – it shows that our methods can be used to drive measurable improvements even in scenarios where agent decisions are noisy.
>
> [1] Vaccaro, M., Almaatouq, A. & Malone, T. When combinations of humans and AI are useful: A systematic review and meta-analysis. Nat Hum Behav 8, 2293–2303 (2024). https://doi.org/10.1038/s41562-024-02024-1

---

> ### Author Response · Authors · 2025-11-20
>
> Upon reflecting more on the reviewers comments, we wonder if they are partly questioning why we didn’t show ILIV-SHAP values by feature alone. The reason for this is that even if ILIV-SHAP tells participants how much human-complementary value the AI gets from each feature, they may still wonder how the AI uses that feature. Therefore ILIV-SHAP is better considered an extension of SHAP. We will make this clear in revising to prevent confusion among readers who might have the same question.

---

### Official Review · Reviewer_C2Li · 2025-10-31

**Soundness:** 2
**Presentation:** 3
**Contribution:** 2
**Rating:** 2
**Confidence:** 4

**Summary:**

The paper attempts to answer the question of whether complementing information improves human-AI collaboration. A preregistered between-subjects experiment with six conditions is performed. A novel instance-level explanation technique is proposed. Together with AI predictions, the new explanation technique appears to help reduce human-AI decision error.

**Strengths:**

The subject of the study is timely in the quick expansion of AI applications.
Hypotheses are tested with a wide choice of experimental conditions.

**Weaknesses:**

The setup of the experiment needs significant improvement.
The claims presented in the paper are inconclusive.

**Questions:**

1. Two fixed features, namely "year remodeled" and "material and finish", are selected as complementing information hidden from the human participants. To make the paper stronger, a varying subset of features of different sizes should be tested.

2. Noise is introduced to the selected complementing attributes to bring down the accuracy of AI1. The authors claim that  "model accuracy differences would not confound our results". Is this true?  Even if both models have similar accuracy, one model may consistently make higher predictions while the other produces random error. Humans may catch the consistency in AI1 and take advantage of it. To clarify, experiments should be set up so that AI1 is trained without noise, with a small amount of noise, and with a large amount of noise. This also helps to understand and quantify the impact of noise on the value of information.

3. Were the participants aware that the highlighted features based on the ILIV-SHAP value are linked to human-complementary information? If so, this may incentivize the participants to always follow the AI as the AI knows more and consequently helps generate higher pay-off.

4. For AI2, instead of using 0k as the SHAP value for features X and Y, a random value should be used. Otherwise, the participants would question the quality of AI and become more skeptical about AI2's predictions.


Minor comments:

On page 3, is there any difference between Pr[w] and pi(w)? Pi is referred to as a probability distribution (on page 3) and then an information model (on page 4). Are these two the same concepts?

---

> ### Author Response · Authors · 2025-11-18
>
> Thank the reviewer for the comments and questions on the experiment. We would first like to note that the controlled experiment (which is the only aspect of our work that the reviewer discusses in their review) is just one part of our overall contribution, with the framework itself being a primary contribution.  We will respond to questions in the order listed by the reviewer.
>
> We designed the study to be intentionally simple and set up complementarity by construction in order to provide a clean proof of concept. If a follow-up paper were to focus exclusively on studying ILIV-SHAP, we agree we would want to vary such aspects of the design.
>
> In our setting, AI1 (the one that has access to full features) has noise in its input while AI2 (the one that only has human-interpretable features) does not. So if the inconsistency will impact the human performance, it should make AI1 has lower human-AI team performance. It sounds like a stronger result when the AI is more noisy but still helps improve human-AI collaboration by having larger ACIV.
>
>
> We informed the participants that the highlighted features are the ones where the AI gets high human-complementary information. We do not agree with the reviewer that this is misleading to the participants: it is a basic description of what they are seeing. By definition, high complementary information means that the AI can perform better than humans by using that feature. Therefore, if they want to achieve higher payoff, humans should incorporate the AI prediction more into their decisions when seeing high ILIV-SHAP values. This is how ILIV-SHAP  is intended to work, and our experiment shows it improves decisions over SHAP as intended.
>
> We disagree with the reviewer that random values should be used for SHAP values of feature X and Y. This yields untruthful explanations, i.e., a feature doesn’t impact the AI prediction but is labeled with a non-zero SHAP value.
>
> In response to the minor comment, $Pr[\omega]$ and $\pi(\omega)$ both denote the prior probability of the payoff-relevant state. $\pi$ is the information model, which can be described by a signal space and a data-generated process. We will remove $Pr[\cdot]$ in the paper to make the notation consistent.

---

> ### Comment · Reviewer_C2Li · 2025-11-21
>
> Thank you for the rebuttal.
>
> I still think the design of the experiment is too simple to address common concerns.
>
> When you add noise to the input to train AI1, you introduce a new dimension of uncertainty in the output. What is the effect of the noise? Does the noise cause errors uniformly in all price ranges or just a few? Do you know exactly whether and how much the "complementing information" has been destroyed, or even enhanced (sounds unlikely but we don't know) as a result? Does it consistently lower the prediction or increase the prediction? How would it impact humans' decisions to go along with AI? Instead of adding noise to the data (input), you could easily add uniform noise to the output of AI1 to bring the accuracy down to the same level as AI2. This way it will be controlled manipulation and the value of "complementing information" will not be tainted.
>
>
> When 0K is presented in the SHAP + AI2 trials, it does not simply suggest "a feature doesn’t impact the AI prediction". It could be interpreted in two different ways: either AI lacks the information (hence is weak) or the information is useless. The former makes AI less trustworthy while the latter encourages the participants to discard the information (X and Y) at their disposal. None of these is a truthful representation of how humans should have used the information to make their predictions. You should simply mark "NA" or use trivial random values instead of 0K throughout the experiment.

---

> > ### Author Response · Authors · 2025-11-21
> >
> > Thank you for the response. Regarding the impacts of the addition of noise to the data, we think it's worth separating concerns about the internal versus external validity of our experimental validation. For our experiment to have internal validity in showing that ILIV-SHAP can be useful, the main requirement is that there be some human complementary information in the AI to explain - the specific effects of adding noise are orthogonal to this. The questions about how adding noise affects the AI prediction and potential complementary information are therefore better described as concerns about external validity - because adding noise changed the nature of the scenario relative to what it would have been with real data, it's not clear how the scenario maps exactly to the real world.
> >
> > Both internal and external validity are important factors in experimental science. However, it is hard to achieve high levels of both in a single experiment. Any controlled experiment will sacrifice some external validity, and vice versa. We tried to strike a balance by starting with real data and showing that we can show measurable improvements in human decision performance rather than simulating human responses, which ML researchers sometimes use in demonstrations. Ultimately we'd like to test ILIV-SHAP with real domain experts in the wild, but such evaluations bring other trade-offs, like lower statistical power  and less of a clean comparison. In this work we focused on providing a rigorously conducted controlled experiment to validate ILIV-SHAP. We again wish to note that ILIV-SHAP is just one component of our paper's contribution, and we demonstrate the value of the framework in several other scenarios as well.
> >
> > Regarding the details of SHAP values, we agree there is a slight semantic difference in interpretation between not including the left out features in the SHAP presentation for AI2 versus including them to show that they had no effect on the model's prediction. It's hard to argue that our inclusion of the features in the explanations was wrong as it accurately conveyed that the model never used those features. More importantly, we think its unlikely that our results hinge on this difference. Regardless of how participants interpreted the role of those features from SHAP, our experiment shows similar human-AI team performance of AI1 and AI2 under SHAP alone, and significantly better human-AI team performance of AI1 than AI2 when ILIV-SHAP and SHAP are combined. However, we can run the setting suggested by the reviewer as a robustness check conditional on the paper being accepted if necessary.

---

### Official Review · Reviewer_3aDh · 2025-11-01

**Soundness:** 3
**Presentation:** 3
**Contribution:** 3
**Rating:** 6
**Confidence:** 3

**Summary:**

The paper considers scenarios where multiple agents make a decision together, as in human-AI decision-making. The authors develop new metrics, ACIV and ILIV, which measure how much informational value the decision of an agent adds to the decision of another. Based on ILIV, they develop a new interpretability method called ILIV-SHAP, which highlights how much each feature contributes to the added informational value.

**Strengths:**

The goal of human-AI systems is to achieve complementary performance and the paper makes an important contribution to this effort, quantifying the complementary value of decisions made by different agents.

The proposed metrics are defined rigorously and presented very clearly. In particular, starting with the conceptually simple ACIV, and then extending it to the instance-wise case, ILIV, and finally incorporating it to SHAP is a good way of organizing the technical parts of the paper (with increasing complexity).

Papers contributing metrics often struggle to provide concrete use cases. However, this paper uses ILIV with SHAP to create a new interpretability method, which is shown to improve human-AI accuracy with human-subject experiments.

The setup for the human-subject experiments is well thought out. The authors have created two versions of the AI agent with the same accuracy but varying ACIV and then observed ILIV-SHAP leads to greater reduction in errors when used with the higher ACIV model. This observation nicely ties all the ideas in the paper together.

**Weaknesses:**

Motivation of the demonstrations in Section 6 is not clear. For instance, experiments for chest X-ray diagnosis and deepfake detection both vary different dimensions (AI models for chest X-rays and features for deepfakes) and report different metrics (ACIVs of combined decisions for chest X-rays and ACIVs of decisions combined with different features for deepfakes). It is not clear why the two settings are set up differently and what hypothesis drove the design of these setups.

What I particularly found hard to understand with these setups: ACIV relative to an empty set is just an expected reward value that doesn’t consider complementarity. Although Figure 2a just reports ACIV relative to empty sets, I can look at the difference between green and orange or green and blue to reason about the complementary information each agent provides to the other. But I was not able to figure out what differences were relevant in Figure 2b.

**Questions:**

ACIV measures how much one decision-maker (D^1) adds information to the decisions of another (D^2) by looking at how the optimal decisions made observing only D^2 differ from the optimal decisions made observing only D^1 and D^2. This seems like a sound measure, especially when the goal is to perform optimally, however isn’t it indirect in the sense that it involves a third decision-maker? What are the challenges associated with trying to quantify the impact of D^1 on D^2 directly?

---

> ### Author Response · Authors · 2025-11-18
>
> Thank the reviewer for the thoughtful comments and questions. Regarding the unclear motivation in the setup of demonstrations in Section 6, we agree more motivation is warranted given that the full information benchmark is defined differently across these. For deepfakes, the signal is high dimensional but the authors of the original study were interested in a smaller set of manually coded features, so we use these plus the different agent decisions together to define the full set of signals on which we define the benchmark against which information value is computed. For the chest X-rays, we were interested in which of multiple AI models would best complement human judgment. We included all of the AI models and the human decisions in computing the full information benchmark to get the best approximation of the value of the features, since each AI model is a garbling of that.
>
> That said, we think a cleaner way to present the framework is to explicitly define the full information benchmark as the best attainable performance given the set of signals that are or will be available to the decision-maker. This targets the framework by default to scenarios where there is a decision-maker with access to some set of features and other agent decisions (such as in the running motivation based in AI-assisted human decision making), and eliminates ambiguity about why one would introduce the notion of a third decision-maker. From this perspective our framework can be viewed as extending the rational agent framework for studying AI reliance from Guo et al (2024) to analyze and design interventions related to complementary information. We will add this new definition to the section on definitions.
>
> Consequently, we updated the radiology example to instead use the radiology image information itself as basic features along with one AI model at a time in creating the full information benchmark. We use a cross-fitted predictor to form the estimator of rational belief while avoiding overfitting of the benchmark. We added the details of cross-validation in the revision (appendix H.2). Similarly, we removed the human-AI decisions from our main presentation of the deepfake example, so that the full information benchmark corresponds to the best attainable performance given the signal structure a human decision-maker facing that problem would have access to.

---

### Author Response · Authors · 2025-11-18

We thank the reviewers for providing their comments and questions. We will reply by points below. We also submitted a revised version of the paper and highlighted the revised content in blue color.

---

### Author Response · Authors · 2025-12-01
**Summary of discussion to AC**

Dear AC,

Thank you for your service especially at this time. We have submitted a revised manuscript to address the reviewers’ questions and actively engaged with all reviewers. We would like to highlight the following regarding the discussion period.

- Reviewer 3aDh acknowledged the contribution of our work but was mainly concerned with the set-up of the demonstrations. We revised the demonstrations to make the set-up aligned and this change doesn’t impact the conclusions.
- Reviewer r9GS listed a number of concerns regarding the design of the experiment, problem framing, and technical details of the algorithm. We addressed their main technical point (we had made a mistake in one figure, which one of their primary critiques concerned) and revised the paper to include the technical details of the algorithm (Reviewer A4pM pointed out the same problem and agreed it is solved). We responded to their other concerns seeking clarification, and noting where we disagreed, as some of their comments described a view of experimental methods that misses our contribution. They did not respond to our rebuttal though we posted it early in the rebuttal period.
- We engaged in a detailed discussion with Reviewer C2Li regarding experimental design and ethics. They did not comment on any of our contributions beyond the proof-of-concept validation for ILIV-SHAP. While we provided clarifications to their follow-up questions, we did not receive a response to our final reply.
- Based on their comments and suggested references, Reviewer A4pM seemed well-acquainted with research in human-AI complementarity. After discussion and revisions based on their comments, they raised their score and explicitly mentioned in their final comment that “early concerns now feel resolved, minor, or acceptable for future work”.

Overall, the review process led to a stronger version of our work. The two reviewers whose comments expressed understanding of our contributions agreed it was an important contribution to studying human-AI complementarity that is worthy of acceptance. Prior to the camera ready, we plan to make the final revision we suggest to ameliorate r9GS's confusion. We are also very much open to running the robustness check we suggested to C2Li if the AC deems is useful, though our personal judgment is that this is overkill.

Best regards,

The Authors

---

### Meta-Review · Area_Chair_d8xD · 2026-01-05

**Summary:**

The paper was reviewed by 4 experts and received scores 2246. The concerns of the reviewers are given below. The AC thinks that a majority of issues were resolved, while some issues still remain. Most of the remaining issues are things that could be fixed with additional experiments (e.g., to test alternative scenarios like different ways to induce noise, or an interface where both side has highlights) or as limitations requiring future work, and minor details/clarifications.  However, the AC thinks that the paper is interesting and theoretically well-founded with experiments that are convincing enough to show the practical value. The authors should consider further updating the paper according to the meta-review, including further experiments / analysis as much as possible in order to make the paper even stronger.

**Reviewer Concerns:**

**Reviewer 3aDh (rating 6)**
1. Motivations for Section 6 is not clear.
2. More intuition about ACIV relative to an empty set?
3. Why not quantify the impact of D^1 and D^2 directly?

The AC thinks Point 1 was addressed. However the authors did attempt to address points 2 and 3 in the response.

**Reviewer C2Li (rating 2)**
1. experiments could be stronger by varying the subset of features (of different sizes).
2. is it the correct way to add noise to AI1?
3. were the participants aware that the highlighted features were based on ILIV-SHAP? It may incentivize participants to always follow the AI.
4. for AI2, use random value fo SHAP values for features X and Y (instead of 0).

Considering the response/discussion:
- For Point 2, the AC thinks that the authors addressed the concern -- adding noise at input is one particular method for inducing the same accuracy in AI1. The reviewer suggests to add noise to the output. There are other approaches also, e.g., subsampling the data. The main concern of the reviewer is that the introduced error in AI1 might be biased (e.g., always adding a constant), which the human could discover. One way to show this is or is not the case is to examine the error induced by the noise on the output of AI1 (the difference between original AI1 and noised AI1), and see if this error is significantly biased. As for other methods to induce noisy outputs, the authors may consider adding this as a limitation of the current study and as directions of future work.
- For Point 3, the AC thinks that the authors addressed the concerns well.
- For Point 4, the authors offer to run additional experiments as a robustness check, which is a good idea.
- For Point 1, it was not addressed by the authors -- the AC thinks that it should be considered as future work to show generalizability of the method.

**Reviewer r9GS (rating 2)**
1. issues with ILIV-SHJAP experiment (RQ2). Baseline SHAP is worse than "no explanation".
2. additive information confound (SHAP is compared with SHAP & ILIV-SHAP)
3. practical utility of ACIV is overstated - poor predictor for magnitude of improvement.
4. weakness of human-AI framing - focus on AI-AI setting?
5. sensitivity to surrogate model A; intractability of ILIV-SHAP.
6. Some claims should be revised (stated in "Questions").

Considering the response/discussion:
- The AC thinks that Points 1 and 2 are addressed well enough, yet the AC agrees with the reviewer that the information presented (SHAP vs SHAP+ILIV) is not equivalent. Perhaps authors could consider another experiment where the least AI-informative features are highlighted -- any difference in result would be directly related to the method used to select the highlighted features, rather than than difference in presentation.
-  The AC thinks that Points 3 and 4 were well addressed.
- Point 5 was actually addressed in response to Reviewer A4pM, who also raised the same issue.
- Point 6 was not discussed by the authors.

**Reviewer A4pM (rating 4)**
1. Revisions of Related Work are needed.
2. Sensitivity analysis of the learned regressors. Safeguards of preventing leakage.
3. flow of theory section needs improvement.
4. computation of SHAP is slow, and potentially biased.
5. limitations of relabeling uninterpretable features (X/Y).
6. what is the interface design?
7. fairness statement in deepfake study.

The AC thinks that all Points were addressed well (as did the reviewer).

**Reviewer Scores:**

The AC thinks that:
- Reviewer A4pM would increase from 4 to 6 or 7 (change to accept).
- Reviewer r9GS's concerns were addressed well enough, except for Point 6, which is a minor concern (easily fixable). Thus the AC thinks that the reviewer would increase to 5 or 6.
- The AC thinks that Reviewer C2Li probably would have raised to 4 or 5, since there was still active discussion and disagreement.
- The AC thinks that Reviewer 3aDh would have kept 6 or increased.

---

### Decision · Program_Chairs · 2026-01-26

Accept (Poster)